# Antrodin C, an NADPH Dependent Metabolism, Encourages Crosstalk between Autophagy and Apoptosis in Lung Carcinoma Cells by Use of an AMPK Inhibition-Independent Blockade of the Akt/mTOR Pathway

**DOI:** 10.3390/molecules24050993

**Published:** 2019-03-12

**Authors:** Hairui Yang, Xu Bai, Henan Zhang, Jingsong Zhang, Yingying Wu, Chuanhong Tang, Yanfang Liu, Yan Yang, Zhendong Liu, Wei Jia, Wenhan Wang

**Affiliations:** 1National Engineering Research Center of Edible Fungi, Key Laboratory of Applied Mycological Resources and Utilization of Ministry of Agriculture, Shanghai Key Laboratory of Agricultural Genetics and Breeding; Institute of Edible Fungi, Shanghai Academy of Agricultural Sciences, Shanghai 201403, China; yanghairui1112@163.com (H.Y.); xubai1994@outlook.com (X.B.); henanhaoyun@126.com (H.Z.); zhangjinsong0313@126.com (J.Z.); wuyingying@saas.sh.cn (Y.W.); tangchuanhong123@163.com (C.T.); aliu-1980@163.com (Y.L.); yangyan@saas.sh.cn (Y.Y.); 2WuXi App Tec Co, Ltd., Shanghai 200131, China; 3College of Life Sciences, Shihezi University, Shihezi 832003, China; 4Food Science College, Tibet Agriculture & Animal Husbandry University, Linzhi 860000, China; liu304418091@126.com

**Keywords:** antrodin C, apoptosis, autophagy, AKT, mTOR, metabolic stability

## Abstract

The current study aims to explore the possible anti-lung carcinoma activity of ADC as well as the underlying mechanisms by which ADC exerts its actions in NSCLC. Findings showed that ADC potently inhibited the viability of SPCA-1, induced apoptosis triggered by ROS, and arrested the cell cycle at the G2/M phase via a P53 signaling pathway. Interestingly, phenomena such as autophagosomes accumulation, conversion of the LC3-I to LC3-II, etc., indicated that autophagy could be activated by ADC. The blockage of autophagy-augmented ADC induced inhibition of cell proliferation, while autophagy activation restored cell death, indicating that autophagy had a protective effect against cell death which was induced by ADC treatment. Meanwhile, ADC treatment suppressed both the Akt/mTOR and AMPK signaling pathways. The joint action of both ADC and the autophagy inhibitor significantly increased the death of SPCA-1. An in vitro phase I metabolic stability assay showed that ADC was highly metabolized in SD rat liver microsomes and moderately metabolized in human liver microsomes, which will assist in predicting the outcomes of clinical pharmacokinetics and toxicity studies. These findings imply that blocking the Akt/mTOR signaling pathway, which was independent of AMPK inhibition, could activate ADC-induced protective autophagy in non-small-cell lung cancer cells.

## 1. Introduction

According to the report, 1.8 million people globally are diagnosed with lung cancer, and 1.6 million people died from this cancer in 2012 [1]. Amongst all cancers, the mortality of lung cancer ranks first and second among men and women, respectively [1]. The most frequent types of lung carcinoma are small-cell lung carcinoma (SCLC) and non-small-cell lung carcinoma (NSCLC), with NSCLC accounting for almost 80% of all lung carcinoma [2,3]. Chemotherapy is one of the major treatments for cancer, although it is known to have strong side effects [4]. Multi-drug resistance and dose limiting adverse reactions are the main reasons for the failure of chemotherapy in NSCLC [5]. Therefore, the development of anti-NSCLC effects and low side effects as well as the safe and long-term use of biologically active substances has become a popular topic of inquiry. Natural bioactive agents, particularly those from medicinal mushrooms, have been considered to have major potential in this field, due to their attractive anticancer effect and hypotoxicity [6,7,8].

Apoptosis is marked by morphological and biochemical changes including chromatin condensation, chromosome DNA fragmentation, global mRNA decay, etc., and is essentially a caspase-dependent proteolysis process [9,10]. Autophagy is a dynamic procedure which involves the bulk degradation of damaged cytoplasmic organelles and proteins via a lysosome-mediated signaling pathway. Excessive autophagy has the potential to lead to type II programmed cell death, while emerging evidence has shown that autophagy may have a cytoprotective effect on malignant cells [11,12,13,14]. It is known that interaction effects between autophagy and apoptosis are complex, sometimes being opposing, and sometimes reinforcing. However, this interaction is key for determining the survival of malignant cells. For this reason, there is a strong possibility that the effect of anti-tumor drugs has a close relationship with the relationship between autophagy and apoptosis.

*Taiwanofungus camphorates* (M.ZangC.H.Su) Sheng H. Wu et al. is a treasured Taiwanese mushroom which only parasitizes in the inner cavity of the endemic species *Cinnamomum kanehirae* Hayata, Lauraceae or the bull camphor tree [15,16]. *Taiwanofungus camphoratus* is known as the ruby in Taiwan’s forest as a result of its excellent biological activities, which include antihepatotoxic, anticancer, anti-inflammatory, antihypertensive, neuroprotective, and antioxidant properties [17,18,19]. In 2016, its anticancer effect was useful for locating antroquinonol, a ubiquinone derivative isolated from the fruiting body of *T. camphoratus*, and was successfully entered into Phase II clinical trials for treating NSCLC due to its excellent anti-lung cancer effect [20]. Antrodin C (ADC), from mycelium of *T. camphoratus,* is a maleimide derivative. According to reports, more than 80% of all bioactive mushroom compounds are isolated from their fruiting bodies. However, compounds from mycelial are considered to have great future potential due to their low cost and a vast market demand [18]. Our preliminary experiments have also shown an anti-tumor effect of ADC on lung cells which was better than for other malignant cells and is similar to the anti-tumor activity of antroquinonol. Metabolic stability has a close relationship with drug clearance, and so candidate compounds for new drugs are in general analyzed in vitro [21]. In vitro stability analysis has the advantages of being relatively low cost and convenient, which can help to reduce the high cost of new drug development [22]. However, there is as yet no literature on the metabolic stability of ADC.

Therefore, our research aimed to ascertain: firstly, whether ADC could inhibit the proliferation of SPCA-1 cells; secondly, whether it is possible to define the precise mechanism of the inhibitory action; and thirdly, to evaluate phase I of the metabolic stability in vitro.

## 2. Results

### 2.1. Effects of ADC In Vitro Cell Proliferation of SPCA-1 and BEAS-2B

The effects of ADC on SPCA-1 cell proliferation were analyzed using alamarBlue^®^. In this study, ADC was incubated with SPCA-1 cells for 72 h, after which the cell proliferation rate was reduced in a dose-dependent manner (Figure 1A). Particularly, at a concentration of 300 μM, ADC treatment could lead to a 71.41% decrease in cell proliferation when compared with untreated cells. The IC_50_ of ADC was 120.14 μM. These results suggest that ADC could demonstrate an inhibitory effect on SPCA-1 cells. 

Low cytotoxicity to normal cells is a key criterion for screening anticancer lead compounds. BEAS-2B cells were isolated from normal human bronchial epithelium as a model system for research of normal human lung epithelium. Therefore, tumor cytotoxicity without damage on normal lung cells was performed by alamarBlue^®^ assay in this study. As shown in Figure 1B, except for 300 uM, the ADC had no inhibition effect on BEAS-2B at 72 h. In this study, the cytotoxicity of ADC to normal cells was very low in vitro. However, cytotoxicity of ADC in vivo needs to be tested in future research.

### 2.2. Effects of ADC In Vitro on the Colony Forming Ability of SPCA-1 Cells

The colony formation experiment was carried out in order to assess cancer cells’ susceptibility and viability in the presence of ADC in an anchorage-independent environment. Results showed that the colony formation ability of SPCA-1 significantly decreased with ADC. As shown in Figure 2, compared with untreated cells, 240 μM of ADC induced a 76% to 50% decrease in the number of colonies, while 75 μM 5-Fu induced a 74% to 32% decrease in the number of colonies. Result indicate that ADC could significantly suppress the susceptibility and viability of SPCA-1 in vitro.

### 2.3. Effects of ADC In Vitro on Cell Migration of SPCA-1 Cells

Migration induced by ADC was analyzed by measuring wound closure in a wound healing experiment. The edge of the wounded area and the wound closure was photographed at 0, 24, and 48 h. When comparing ADC treatment, a remarkable increase in wound closure was observed in the untreated cells. As shown in Figure 3, following treatment with ADC, wound closure barely moved, whereas in the untreated cells, the wound closure distance shortened 0.17 times and 0.13 times when treated with 150 μM ADC for 24 and 48 h, respectively. This result suggested that ADC can significantly reduce the cell migration ability of SPCA-1 cells.

It is well known that the proliferation of cells is closely related to their migration ability. The proliferation of the epithelial cells was shown to be the primary force that drives this cell migration along the villus. Passive mitotic pressure generated by cell division in the intestinal crypts as well as the subsequent gradual expansion in cell diameter along the crypt–villus axis provides a plausible explanation for the steady, continuous migration of epithelial cells [23,24]. Indeed, previous computational models have suggested that these forces alone are sufficient to explain observed rates of cell migration, at least within the crypt [25,26,27,28,29,30]. Matrix metalloproteinases (MMPs) are a family of zinc-dependent extracellular matrix (ECM) re-modelling endopeptidases which have the ability to degrade almost all components of an extracellular matrix and are implicated in various physiological as well as pathological processes [31]. Carcinogenesis is a multi-stage process in which alteration of the microenvironment is required for the conversion of normal tissue to a tumor [32]. Initially, MMPs were considered to be important only in invasion and metastasis. However, recent studies have shown that MMPs are involved in several steps, such as proliferation, apoptosis, and angiogenesis during carcinogenesis [33]. 

As shown in Figure 4A, ADC treatment for 24 h led to a decrease in MMP-9 expression when compared with the negative control. As is already known, ginkgolide C (GC) is an MMP-9 activator. This was used to explore the exact effect of MMP on proliferation and migration of SPCA-1. As shown in Figure 4B, the proliferation rate of SPCA-1 increased by the combination treatment with 75 µM ADC and GC (10 and 20 µM) when compared with treatment of 150 µM ADC alone. We also found that co-incubation of 75 µM ADC and 10 µM GC with SPCA-1 could attenuate the inhibition effect of ADC on migration ability (Figure 4C,D). Taken together, the above results suggest that MMP-9 played a key role on both the proliferation and migration abilities of SPCA-1.

### 2.4. Influence of ADC on Cell Cycle of SPCA-1 Cells

Analysis of the cell cycle based on PI staining and quantitative detection of cells were both performed using fluorescence-activated cell sorter (FCAS). As shown in Figure 5, following treatment with a gradient concentration of ADC (37.5, 75, and 150 μM), the percentage of cell cycles at the S phase rose from 36.87% to 43.66%, while at the G2/M phase, this decreased from 11.85% to 5.11%. This suggested that the S cell cycle arrest, which was induced by ADC, led to a further suppression of cell proliferation.

### 2.5. Effects of ADC on SPCA-1 Cell Apoptosis

In order to assess the effects of ADC on the apoptosis of lung cancer cells, SPCA-1 cells were exposed to 37.5, 150, and 240 μM ADC for 72 h. A FACS experiment was performed. As can been seen in Figure 6, ADC treatment resulted in an increased proportion of early apoptotic cells and late apoptotic/necrotic cells when compared with the negative control. The total impaired rate of cells was increased by 93% when using 240 μM ADC compared with untreated cells. This result suggests that ADC could have a significant effect on SPCA-1 apoptosis.

### 2.6. Effects of ADC on ROS Released by SPCA-1 Cells

Although there is no clear evidence of its origin of reactive oxygen species (ROS), ROS was identified as being involved in cell apoptosis as a crucial intermediate stress response [34]. In this study, DCFH-DA was chosen to demonstrate whether apoptosis of SPCA-1 was induced by ADC through ROS generation. We found that ADC can increase ROS production at a concentration of 30 μM, 45 μM, and 60 μM (Figure 7A), and that ROS production increased 8.6%, 14.9%, and 23.1%, respectively (Figure 7B). This result suggested that ADC had the potential to induce proliferation inhibition and apoptosis of SPCA-1 through an ROS release. As we know, N-acetyl-L-cysteine (NAC) is an ROS scavenger. In this study, NAC was used to determine which ROS was involved in the ADC-induced proliferation inhibition as well as the apoptosis of SPCA-1. As can be seen in Figure 7C, the ADC-induced inhibition effect on cell proliferation was clearly attenuated by NAC. Results shown in Figure 7D,E demonstrate rates of early apoptosis, and total impaired cells were decreased following combination treatment with ADC and NAC when compared with just the ADC treatment.

### 2.7. Effects of ADC on Activation of Caspase-3, P53, and Bcl-2

Caspases are the important intermediate of apoptosis. Once the proteolytic is activated, procaspase forms will be converted to their enzymatically active forms. In addition, the consecutive activation of caspases is the crucial meditator of the mitochondrial apoptotic pathway [35]. Once SPCA-1 cells were treated with 100 μM ADC for 72 h, cleaved caspase-3 expression reached up to 15.33-fold. P53, known as a genes transregulator, is involved in DNA synthesis, repair, and apoptosis. When compared with untreated cells, the co-culture of SPCA-1 cells with 100 μM ADC caused P53 expression of up to 3.31-fold. In addition, Bcl-2 expression was down 1.45-fold after being treated with 100 μM ADC for 72 h compared with untreated cells. These results (Figure 8) suggest that caspases and the Bcl-2 family have taken part in ADC-induced SPCA-1 cell apoptosis by increasing the expression of pro-apoptotic members (cleaved caspase-3 and P53) and decreasing the expression of anti-apoptotic members (Bcl-2).

### 2.8. Effects of ADC on Autophagy Activation in SPCA-1 Cells

Autophagy has been of great interest for those developing novel anticancer treatments in recent years. In order to testify whether ADC induced autophagy activation of SPCA-1, TEM was employed to detect the formation of autophagic vacuoles. Autophagic vacuoles formation, which contain lamellar structures or residual digested material as well as empty vacuoles, is an indication of autophagy activation. Increased numbers of vacuoles and mature autophagosomes were frequently observed in cells when they were treated with 200 μM ADC, which indicated that ADC could activate autophagy of SPCA-1 (Figure 9A).

The conservation of LC3-I to LC3-II through lysis, lipoification, and proteolysis is a crucial character of autophagy activation, and so the LC3 protein serves as one of the most important markers of autophagy. Therefore, it is necessary to confirm whether ADC treatment could induce redistribution of LC3 (autophagosome) in SPCA-1. Results of the FCAS indicated that ADC could activate LC3-II formation (Figure 9B). Western blotting results also demonstrated that conversion of LC3-I to LC3-II increased 479.23%, 403.19%, and 382.27% following 50, 100, and 200 µM ADC treatment when compared with untreated cells (Figure 9C,D). The mRFP-GFP-LC3 detected by the live-cell imaging method was used to observe the flux rate of autophagy of SPCA-1 cells, while the reduction of GFP was used to indicate the process of autophagy. Results demonstrated that expressed GFP-LC3 was downregulated by ADC treatment, dependent on dose, while a combination of RFP and GFP was detected as yellow speckles in the untreated cells more than in the ADC treated cells, suggesting that autophagy was induced through activation of autophagic (Figure 9).

As we know, both lysosome and autophagosome formation have the potential to induce an increased level of LC3B-II. However, in order to identify the reason for increased levels of LC3-II, the late-stage autophagy inhibitor chloroquine (CQ, 100 μM) must be used. Results indicated that LC3-II expression in SPCA-1 cells was increased 59.02%, 75.73%, and 21.15% following the combination treatment with 50, 100, and 200 µM ADC when compared with sole treatment of ADC, which suggests that increased levels of LC3B-II were mainly due to the increase in the number of vacuoles and mature autophagosomes (Figure 10A,B). In summary, these results demonstrate that autophagic activity (autophagic flux) was upregulated through the increased level of autophagosome formation in SPCA-1 cells which were treated with ADC.

### 2.9. Protective Effects of Autophagy on ADC-Induced Apoptotic SPCA-1 Cell Death 

Autophagy has the potential to either protect cell survival by serving as an antagonist to block apoptotic cell death or by contributing to cell death [36]. Therefore, the exact role of autophagy in the anti-NSCLC activity of ADC was clarified in this study. A significant decrease (18.74%) or increase (17.08%) was found in the growth rate of SPCA-1, of which autophagy was inhibited by CQ or activated by RAPA following ADC treatment, in contrast with treatment by ADC alone (Figure 11A). Cell viability data was similar to results found in cell apoptosis. As shown in Figure 11C,D, apoptotic cells were increased 1-fold by combination treatment with 150 µM ADC and CQ when compared with treatment of 150 µM ADC alone. These results indicated that ADC induced protective autophagy in SPCA-1 cells. In this study, the RAPA was further used to explore the extract stage of protective autophagy, and the results showed that pre-treatment with RAPA did not restore the ADC-triggered cell death (Figure 11B). When combined with the results of RAPA in Figure 11A,B, this suggests that the survival effect only occurred when the cells were under stress. 

### 2.10. ADC Downregulated the AKT-mTOR Pathway and AMP-Activated Protein Kinase (AMPK) Pathway

As we know, AMPK and PI3K/Akt/mTOR are the two key signaling pathways which regulate autophagy and apoptosis. Recent reports have suggested that autophagy is negatively regulated by the PI3K/Akt/mTOR pathway [37]. To determine whether ADC-induced autophagy is involved in the Akt-mTOR pathway, FCAS were used for further detection. A phosflow analysis demonstrated that the phosphorylation of mTOR (Ser2448) and Akt (Ser473) was downregulated by ADC treatment in SPCA-1 cells, dependent on dose, which suggested that ADC-induced autophagy was activated through downregulation of the Akt-mTOR pathway (Figure 12A). 

AMPK is a key signal factor involved in regulating the balance of cellular energy. When the cell is hungry, the AMPK signaling pathway will tell the body to increase glucose and fatty acid uptake. Recently, an increasing number of studies have shown that the AMPK signaling pathway is related to autophagy activation [37]. Western blotting results have indicated that the relative expression of pAMP/AMPK decreased by 17.67%, 37.63%, and 34.20% following 50, 100, and 200 µM ADC treatment, respectively, when compared with untreated cells (Figure 12B,C). Based on the results above, autophagy induced by ADC has a close relationship with the AMPK inhibition-independent blockade of the Akt/mTOR signaling pathway in SPCA-1 cells. 

### 2.11. Metabolic Stability of ADC in SD Rat and Human Liver Microsomes

As shown in Table 1, following incubation with rat liver microsomes for 0, 5, 10, 20, 30, and 60 min, results were reported as percent remaining for ADC, which was 100%, 52.5%, 30.2%, 13.3%, 7.7%, and 4.6%, respectively (Figure 13). The value of T_1/2_ (min) in the rat liver microsomes, which was calculated by the % of remaining data versus time, was 7.5 (Table 1). The values of CL_int(mic)_ (μL/min/mg protein) and CL_int(liver)_ (mL/min/kg) in the rat liver microsomes were 185.8 and 334.4, respectively (see Table 1, above).

As can be seen in Table 1, following incubation with human liver microsomes for 0, 5, 10, 20, 30, and 60 min, the percent remaining for the ADC was 100%, 92.5%, 85.8%, 72.2%, 68.0%, and 47.8% respectively (Figure 13). The value of T_1/2_ (min) in the human liver microsomes was 54.1 (Table 1). The value of CL_int(mic)_ (μL/min/mg protein) and CL_int(liver)_ (mL/min/kg) in the human liver microsome were 25.6 and 23.0, respectively (Table 1).

## 3. Discussion

A pressing problem in the success of NSCLC therapy is tackling the diverse side effects and resistance of chemotherapy to current anticancer agents. Therefore, new strategies for drug development are urgently needed to solve the problem of drug resistance in anti-NSCLC treatment. In recent years, medicinal mushrooms have obtained global attention as a result of their strong therapeutic activity as both chemical inhibitors and immunomodulators.

Previous studies have reported that the ethanolic extract of *T. camphoratus* treatment could induce cell cycle arrest and proliferation inhibition of Hep3B and HepG5 [38]. Chen and colleagues [39] have found that both incubation amphotericin B and *T. camphoratus* could induce significant increases in proliferation inhibition and apoptosis rates of HT29 cells. The noteworthy anticancer-effect of *T. camphoratus* may contribute to the large amounts of polysaccharides, terpenoids, succinic acid, and maleic acid derivatives. According to existing research, antroquinonol has excellent anti-tumor effects on various malignant cells [40]. In this study, ADC exhibited anticancer effects on human NSCLC cell lines SPCA-1. These effects included cell growth inhibition, cell migration reduction, ROS release, cell cycle arrest, autophagy, and apoptosis. The effective G2 checkpoint mechanism is a key characteristic of many malignant cells. Based on these findings, abrogation of the cell cycle G2 checkpoint may become a new target for anti-NSCLC chemotherapy [41]. It has been found that the anti-tumor target of several lead compounds was the G2/M cycle arrest-dependent inhibition of cell proliferation [42,43]. In this study, the form of the cell proliferation inhibition induced by ADC has been shown to be dependent on the G2 cycle arrest, a finding which indicated that ADC can be used as a potential anti-NSCLC agent.

Inducing the apoptosis of malignant cells is one of the most important strategies for cancer chemotherapy. Therefore, the effect of ADC on NSCLC apoptosis was assessed in this study. The result of the FCAS experiment showed that ADC treatment could significantly increase the number of apoptotic cells, which is positively correlated with its concentration, and so suggested that ADC exhibited anticancer activity through apoptosis. Reactive oxygen species is a required metabolite of normal metabolism in cell growth which plays a critical role in the stress response as an important mediator. Some studies have shown that excessive ROS led to the occurrence of apoptosis [44,45,46]. The FCAS results showed that ADC induced ROS release, which could be a factor of the occurrence of apoptosis. The similar ROS-mediated cytotoxicity of other anti-NSCLC compounds has been found by other researchers. Zhou et al. [47] found that luteoloside could induce proliferation inhibition of NSCLC through ROS-mediated G_0_/G_1_ arrest. Song and colleagues [48] also discovered that the anti-tumor effect of SZC017, an oleanolic acid derivative, had a close relationship with ROS-dependent apoptosis. As we know, the loss of mitochondrial transmembrane potential will lead to ROS generation. The results of Zhang et al. [49] demonstrated that cedrol treatment could lead the loss of mitochondrial transmembrane potential and cause ROS-mediated apoptotic cell death in A549 NSCLC. The above results have indicated that the anti-tumor effects of these compounds were a close relationship with the AKT-related signaling pathway. Interestingly, Wang et al. [50] demonstrated that the role of ROS on endothelial dysfunction was opposite to its role on NSCLC. Their results showed that ADC could inhibit hyperglycemia-induced endothelial cell dysfunction via the inhibition of ROS generation, senescence, growth arrest, and apoptosis in cultured HUVECs [50]. In addition, Wang and colleagues [50] found that the Nrf2-related signaling pathway was involved in the protective effects of ADC on hyperglycemia-induced vascular endothelial cells’ senescence and apoptosis. Therefore, the different signaling pathways in the two diseases may be the main reason for the inconsistency of ROS roles.

The Bcl-2-related protein family is the most representative apoptotic regulatory gene family [51]. Chaetoglobosin K25 [52] and buforin IIb 26 [53] have been found to activate apoptosis through a decrease of Bcl-2 expression. The Bcl-2-related protein family consists of a pro-apoptotic gene (*Bcl-2, Bcl-xL, etc.*) and an anti-apoptotic gene (*Bax, Bak, etc.*). The dialogue between the pro-apoptotic gene and the anti-apoptotic gene will determine the fate of a malignant cell [54].

Moreover, P53 plays a diverse role in Bcl-2-dependent apoptosis which substantially affects the ADC-mediated cell death. The proapoptotic protein Bax is activated by P53, while the anti-apoptotic protein Bcl-2 expression is decreased by P53 [55]. Once pro-apoptotic factors are released, the caspase family was further activated, resulting in caspase-3 cleavage. Our results indicated that ADC-induced apoptosis could be attributed to the decrease of Bcl-2 protein and increase of caspase-3 and P53 proteins in the SPCA-1 cells.

The role of autophagy in malignancy has attracted much attention due to the viable option it offers cancer therapies. Our results showed that ADC-induced autophagy promoted the formation of autophagosomes, activated the conversion of LC3B-I to LC3B-II, and induced autophagy flux.

The relationship between cancer and autophagy is complex and hard to unpick. According to the literature, autophagy activators are used clinically for anticancer treatment [56]. However, in contrast to the positive effects of autophagy activators on apoptosis-resistant malignancy treatment, some researchers have found that drug resistance in some malignant tumors might be the result of a relationship with autophagy activation [57]. Autophagy can inhibit tumor genesis by maintaining cell stability and reducing cell damage [58]. The apoptosis sensitivity of cancer cells can be increased by inhibiting autophagy, particularly in the late stage [59]. In this study, autophagy inhibitors CQ and autophagy activators RAPA were used to determine the exact effects of altered autophagy on NSCLC death, induced by ADC. In accord with these findings, the results of our study indicated that autophagy inhibition could accelerate ADC-induced cell death, and that autophagy activation could restore ADC-induced cell death. Furthermore, the combination of pro-apoptosis agents and anti-autophagy agents is perhaps a magic bullet in accelerating the anti-tumor effect on human NSCLC.

As we know, the PI3K/Akt/mTOR signaling pathway plays an important role on migration, invasion, and proliferation of malignant tumors. Duan and colleagues [60] found that the PI3K/ATK/mTOR pathway was related to invasion, migration, and proliferation of colorectal cancer induced by IMPDH2. Paul et al. [61] demonstrated that the binding between SDS22 (protein phosphatase 1 regulatory subunit) and AKT has the potential to lead to dephosphorylate ATK at Thr308 and Ser473 through PP1, and hence can inhibit proliferation, invasion, and migration of tumor cells. These findings indicate that the inhibition of the Akt/PI3K/mTOR pathway might lead to proliferation, migration, and invasion of SPCA-1. Studies have confirmed that many compounds suppress tumor growth and activate autophagy through the downregulation of this pathway. The autophagy activation of Rotten was found in prostate and pancreatic cancer stem cells, and results indicated this activation effect was dependent on the Akt/PI3K/mTOR pathway [62,63]. As shown in our results, the phosphorylation of mTOR and Akt was dose-dependently downregulated by the ADC treatment, thus initiating autophagy in SPCA-1 cells. Results implied that inhibition of the mTOR-Akt pathway was related to autophagy, which was induced by ADC autophagy. 

As well Akt, AMPK, another important mTOR regulator, is activated in intracellular and external environmental pressures as a crucial energy sensor [64]. Studies have shown that AMPK participates in controlling the fate of cancer cells [65]. That is, AMPK activation inhibited mTOR both in the cell cycle arrest and apoptosis in vitro and in vivo. In contrast, cells were protected by the AMPK signaling pathway from chemotherapy-induced apoptosis and metabolic stress in certain circumstances [66]. The mTOR-mediated autophagy in cancer cells seems to be impacted both by AMPK-dependent cytotoxicity and cytoprotection [67,68,69,70]. Our results have shown that phosphorylation of AMPK and AMPK was dose-dependently downregulated by the ADC treatment, which indicates that ADC-induced autophagy was independent of AMPK inhibition while being mediated via the inhibition of the Akt/mTOR signaling pathway. 

The microsomal stability assay is widely used in vitro model to characterize the metabolic conversion by phase I enzymes, in which the cytochrome P450 (CYP) family are the most important. Since metabolism is known to be highly variable in different species, microsome stability assay is commonly run in multiple species. The most commonly used species include humans for predicting clinical pharmacokinetics, rats for toxicity studies, mice for efficacy models, and dogs and monkeys for large animal toxicity. In this study, we tested the metabolism of ADC in rat and human liver microsomes. Results demonstrated that ADC was highly metabolized in SD rat liver microsomes, and moderately metabolized in human liver microsomes. This suggests the need for further modification of the metabolic stability of ADC and also contributes to predicting the in vivo PK performance [71,72].

## 4. Materials and Methods 

### 4.1. Chemicals and Reagents

*T. camphoratus* (Access number: J1) was from the Preservation Center of Fungi, Institute of Edible Fungi, Shanghai Academy of Agricultural Sciences (Shanghai, China). Human lung adenocarcinoma cell line SPCA-1 (Access number: TCHu 53) was purchased from Cell Resource Center of Shanghai Institute of Life Sciences, Chinese Academy of Sciences (Shanghai, China). The BEAS-2B (Acess number: GDC0139) was purchased from China Center for Type Culture Collection (Wuhan, China). Ribonuclease A (RNase A, #EN053), Pierce^TM^ BCA protein assay kit (#23225), Dulbecco’s Modified Eagle Medium (DMEM, #11965-092), no phenol red DMEM (#31053-028) and fetal bovine serum (FBS, #10099-141) were from Thermo Fisher Scietific Inc. (Waltham, MA, USA). Annexin V-fluoresceinisothiocyanate (FITC) apoptosis detection kit was from Nanjing KeyGen Biotech. Co. Ltd. (Nanjing, China). N-acetyl-L-cysteine (NAC), Chloroquine (CQ, #PHR1258), 5-Fluorouracil (5-Fu, #03738) and Propidium iodide (PI, #P4170) were from Sigma–Aldrich Co. LLC (St Louis, MO, USA). alamarBlue^®^ (#BUF012B) was from Bio-Rad Laboratories, Inc. (Hercules, CA, USA). The Bcl-2 ELISA assay kit, P53 ELISA assay kit, and caspase-3 (Asp175) DuoSet IC ELISA were from R&D Systems Inc. (Minneapolis, MN, USA). Ginkgolide C(GC), rapamycin (RAPA) and chloroquine (CQ) were purchased from Sigma–Aldrich (St Louis, MO, USA). The Cyto-ID^®^ Autophagy detection kit was obtained from Enzo Life Sciences Inc. (Farmingdale, NY, USA). The polyvinylidene difluoride (PVDF) membrane was purchased from EMD Millipore Inc. (Billerica, MA, USA). Phospho-PI3 kinase p85 (Tyr458)/p55 (Tyr199) Antibody was from Cell Signaling Technology (Beverly, MA, USA). The LC3 antibody was from Abcam Co. (Cambridge, UK). The Human liver microsomes (#452117) were from BD Gentest (Corning, NY, USA), and the rat liver microsomes (#R1000) were from Xenotech (Kansas, KS, USA). RayBio^®^ Human MMP-9 ELISA kit (#P14780) was from RayBiotech, Inc. (Norcross, GA, USA).

### 4.2. Cell Lines and Cell Culture

The SPCA-1 and BEAS-2B were cultured in DMEM or DMEM/F12 supplemented with 10% FBS, 1% penicillin–streptomycin at 37 °C and 5% CO_2_ in an incubator with a humidified atmosphere.

### 4.3. Cell Viability Assay

The SPCA-1 or BEAS-2B were seeded in 96-well plates at 2 × 10^4^ cells/mL and cultured overnight. Cells were then treated with ADC (4.69, 9.38, 18.75, 37.5, 75, 150, and 300 μM) for 72 h. Medium containing 5‰ DMSO and 5-Fu (100 μM) served as negative and positive controls. After treatment, the medium was aspirated, and fresh phenol-red-free medium containing 5 μg/mL alamarBlue^@^ was added into all tested and control wells. After incubation at 37 °C for 4–6 h, when the medium color changed, the absorbance at 570 nm and 600 nm was measured using a spectrophotometric plate reader (Bio-Tek Instruments, Inc, Winooski, VT, USA). The proliferation rate was calculated according to the following formula:$$\mathrm{Proliferation}\;\mathrm{rate}\;\left(\%\right)\;=\;1\;-\;\frac{117216\;\times \;A_{570\;(\mathrm{sample})}\;-\;80586\;\times \;A_{600\;(\mathrm{sample})}}{117216\;\times \;A_{570\;(\mathrm{control}}{}_{)}\;-\;80586\;\times \;A_{600\;(\mathrm{control})}}\;\times \;100\%$$

### 4.4. Clone Formation Assay

Cells (2 × 10^5^ cells/mL) were seeded in a 12-well plate and cultured overnight. The cells were then treated with 37.5, 150, and 240 μM ADC for 72 h. Medium containing 5‰ DMSO and 5-Fu (30, 60, and 75 μM) were served as the negative and positive controls, respectively. Cells were then seeded in 6-well culture plates at 100 cells/well. Each treatment had three repeated wells. After incubation for two weeks at 37 °C, cells were washed twice with PBS, and fixed with 40 μL/mL paraformaldehyde, and then stained with crystal violet. Groups of 50 or more cells were counted as colonies. Clone formation efficiency was calculated as (number of colonies/number of cells inoculated) × 100%.

### 4.5. Wound-Healing Assay

Cells were seeded in 12-well plates at 2 × 10^5^ cells/mL and cultured overnight. A scrape was placed through the middle of the confluent cultures with a sterile pipette tip, and washed with PBS to remove debris, followed by treated with 37.5, 75, and 150 μM ADC. Medium containing 5‰ DMSO (5 μL/mL) and 200 μM 5-Fu were served as the negative and positive controls, respectively. The wound was observed under a phase-contrast microscope everyday (Olympus Corporation, Tokyo, Japan).

To further verify the role of MPP-9 on proliferation and migration of SPCA-1, an MMP-9 ELISA assay kit and MMP-9 activator GC were used. Cells (2 × 10^5^ cells/mL) were seeded in 12-well plate and cultured overnight. The cells were then treated with 37.5, 75 and 150 μM ADC for 72 h. Medium containing 5‰ DMSO and 100 μM 5-Fu were served as the negative and positive controls, respectively. After treatment, cells were collected and washed with cold PBS, and resuspended in RIPA lysis buffer, and then left on ice for 30 min. The lysate was centrifuged at 1 × 10^4^ rpm at 4 °C for 30 min, and then total protein level was measured with a BCA assay kit. The absorbance was determined by a spectrophotometric plate reader at 450 nm. The samples (100 µL) were added to each well of the MMP-9 plate, and the plate was covered with a sealer. After incubation for 2.5 h at room temperature with a gentle shake, the plate was washed with 300 µL wash buffer four times. After the last wash, any remaining wash buffer was completely removed, and then the plate was inverted and blotted against clean paper towels. The 1× prepared biotinylated antibody was added to each well of the plate, and then the plate was incubated for 1 hr at room temperature with gentle shaking. The solution was dicarded, and the wash step was repeated. The prepared streptavidin solution was added to each well, and then plate was incubated for 45 min at room temperature with gentle shaking. The solution was dicarded, and the wash step was repeated. The TMB one-step substrate reagent was added to each well, and then the plate was incubated for 30 min at room temperature in the dark with gentle shaking. The stop solution (50 µL) was added to each well, and then the absorbance was determined by a spectrophotometric plate reader at 450 nm. 

The SPCA-1 were seeded in 96-well plates at 2 × 10^4^ cells/mL and cultured overnight. Cells were then treated with GC (5, 10, and 20 μM), ADC (75 μM) or ADC (75 μM) + GC (5, 10, and 20 μM) for 72 h, respectively. The proliferation was analyzed by alamarBlue^@^ assay as shown in Section 4.3.

Cells were seeded in 12-well plates at 2 × 10^5^ cells/mL and cultured overnight. A scrape was placed through the middle of the confluent cultures with a sterile pipette tip, and washed with PBS to remove debris, followed by treated with 10 μM GC, 75 μM ADC or 75 μM ADC + 10 μM GC for 48h, respectively. Medium containing 5‰ DMSO (5 μL/mL) was served as the negative control. The wound was observed under a phase-contrast microscope everyday (Olympus Corporation, Tokyo, Japan).

### 4.6. Cell Cycle Analysis

Cells were seeded in 12-well plates (2 × 10^5^ cells/mL) and cultured overnight. The cells were treated with 37.5, 75, and 150 μM ADC for 72 h. Medium containing 5‰ DMSO and 5-Fu (100 μM) were served as the negative and positive controls, respectively. After treatments, the cells were harvested, washed with PBS, and fixed in 70% ethanol at 4 °C overnight. Cells were collected by centralization, washed with PBS twice, and added with 100 μL PBS containing 0.01 μg/mL PI and 0.005 mg/mL RNase A, then incubated at room temperature in dark for 15 min. Cells were analyzed using a fluorescence activated cell sorting (FCAS), and the results were analyzed using Modfit software.

### 4.7. Cell Apoptosis Detection

Cell apoptosis was determined by Annexin V-FITC apoptosis detection kit according to the manufacturer’s instruction. Briefly, cells (2 × 10^5^ cells/mL) were seeded in 12-well plate and cultured overnight. The cells were then treated with 37.5, 150, and 240 μM ADC for 72 h. Medium containing 5‰ DMSO and 5-Fu (100 and 400 μM) served as the negative and positive controls, respectively. After different treatments, cells were harvested and washed twice with cold PBS, and then resuspended in 100 μL 1× binding buffer solution containing 0.2 μg/mL Annexin V-FITC and 0.5 μg/mL PI, and then incubated at room temperature for 10 min in the dark. Apoptotic cells were analyzed using BD Accuri C6 flow cytometer (BD Biosciences, San Jose, CA, USA).

### 4.8. ROS Detection

2′,7′-Dichlorofluorescin diacetate (H_2_DCFDA) is a cell permeable non-fluorescent probe. Upon oxidation by any hydroperoxide such as ROS, the non-fluorescent H_2_DCFDA is converted to the highly fluorescent 7′-dichlorofluorescein (DCF). Fluorescent density reflected ROS level [73]. Therefore, intracellular ROS levels were determined using an H_2_DCFDA probe as described. Briefly, cells were seeded in 24-well plates (2 × 10^4^ cells/mL) and cultured overnight. Medium containing 5‰ DMSO served as the negative controls. After treatment with ADC (30, 45, and 60 μM) for 24 h in phenol-red-free DMEM, cells were incubated with 25 μM H_2_DCFDA for 30 min at 37 °C. Then cells were harvested, washed, and resuspended in PBS. Cells were analyzed using FCAS, and the results were analyzed using Flowjo software.

To further verify the role of ROS in ADC-induced inhibition of SPCA-1 proliferation, a ROS scavenger was used to treat the cells before ADC treatment. The SPCA-1 cells were seeded in 96-well plates at 2 × 10^4^ cells/mL and cultured overnight. Cells were pretreated with 1, 2, and 4 mM NAC for 2 h prior to exposure to 200 μM ADC for 72 h. The proliferation was analyzed by alamarBlue^@^ assay as shown in Section 4.3. 

To further verify the role of ROS in ADC-induced apoptosis of SPCA-1, NAC was used to treat the cells before ADC treatment. The SPCA-1 cells were seeded in 12-well plates at 2 × 10^5^ cells/mL and cultured overnight. Cells were pre-treated with 1, 2, and 4 mM NAC for 2 h prior to exposure to 150 μM ADC for 72 h. The apoptotic cells were analyzed by FCAS after annexin V-FITC/PI double staining as shown in Section 4.7. 

### 4.9. ELISA Assay

Cleaved caspase-3, Bcl-2, and P53 protein expression were measured using commercial cleaved caspase-3, Bcl-2, and P53 assay kits according to the manufacturer’s instructions, respectively. Briefly, cells (2 × 10^5^ cells/mL) were seeded in 12-well plates and cultured overnight. The cells were then treated with 150 μM ADC for 72 h. Medium containing 5‰ DMSO and 100 μM 5-Fu served as the negative and positive controls, respectively. After treatment, cells were collected and washed with cold PBS and resuspended in RIPA lysis buffer, and then left on ice for 30 min. The lysate was centrifuged at 1 × 10^4^ rpm at 4 °C for 30 min, and then total protein level was measured with a BCA assay kit. The absorbance was determined by a spectrophotometric plate reader at 450 nm. The sample or standard (100 µL) were added to each well of the plate of cleaved caspase-3, Bcl-2 or P53, and the plates were covered with a sealer. After incubation for 2 h at room temperature, the plates were washed with wash butter three times. A diluted detection antibody (100 µL) was added to each well, and the plates were covered with a new plate sealer. After incubation for 2 h at room temperature, the plates were washed with wash butter three times again. The diluted Streptavidin-HRP A was added to each well of the plates, and the plates were incubated for 20 min in dark. After incubation for 20 min, the stop solution was added to each well, and the plates were gently tapped to ensure thorough mixing. The absorbance was determined by a spectrophotometric plate reader at 450 nm. The increase in protein expression was calculated as the ratio between values obtained from the treated samples versus those obtained in untreated controls.

### 4.10. Transmission Electron Microscopy Analysis

Transmission electron microscopy (TEM) was employed to identify autophagosomes in lung adenocarcinoma cell line SPCA-1. Cells were treated with 200 μM ADC for 72 h. Cells were harvested, washed and fixed in 2.5% glutaraldehyde in 0.1 M phosphate buffer, then post-fixed in 1% osmium tetroxide buffer. After dehydration in a graded series of ethanol, the cells were embedded in spur resin. Thin sections (70 nm) were cut on an ultramicrotome. The sectioned grids were stained with saturated solutions of uranyl acetate and lead citrate. The sections were examined by a JEM1230 electron microscope from JEOL (Tokyo, Japan).

### 4.11. Flow Cytometric Analysis of Autophagy with Cyto-ID Staining

The Cyto-ID autophagy detection kit (ENZ-51031-K200) from Enzo Life Sciences (New York, NY, USA) was used to stain live cells according to the manufacturer’s instructions. This assay is dependent on a 488 nm-excitable green fluorescent detection reagent, which specifically fluoresces in autophagic vesicles [73]. Cells (2 × 10^5^ cells/mL) were seeded in 12-well plates and cultured overnight. The cells were then treated with 50, 150, and 200 μM ADC for 72 h. Medium containing 5‰ DMSO and combination of 1 μM CQ or 50 0nM rapamycin served as the negative or positive controls, respectively. After treatment, cells were washed with DPBS, and then incubated with Cyto-ID Green containing indicator free cell culture medium containing FBS (100 μL/mL) for 30 min at 37 °C, 5% CO_2_ in the dark. At the end of staining procedure, the Cyto-ID Green containing medium were washed away, and then cells were trypsinized, washed, and resuspended in ice-cold DPBS containing FBS (40 μL/mL). Cells were analyzed using FCAS, and the results were analyzed using Flowjo software.

### 4.12. Western Blotting Analysis

Whole cell lysates were extracted using RIPA lysis buffer containing protease inhibitor and phosphatase inhibitor. Protein concentrations were measured by a Pierce BCA protein assay kit. Proteins were separated using SDS-PAGE and transferred to PVDF membranes and subsequently blocked with 5% BSA, and incubated with primary antibody (1:1000) from Cell Signaling Technology (Danvers, MA, USA) overnight at 4 °C. Immunoblots were followed by 1–2 h incubation in secondary antibody (1:300, BOSTER). After washing three times with TBS-T, the signal was detected using TanonTM High-sig ECL Western Blotting Substrate from Tanon Science & Technology Co., Ltd. (Shanghai, China) by the Amersham Imager 600 from GE Healthcare (Pittsburgh, PA, USA).

### 4.13. Live-Cell Imaging for Autophagic Flux

Laser scanning confocal microscopy were employed to identify autophagic flux. The SPCA-1 cells were transfected with mRFP-GFP-LC3B-expressing plasmid by Turbofectmin, and then cultured for 6 h. Cells were treated with 20, 150, and 200 μM ADC for 24 h, followed by incubation with DAPI for 5 min, cells were washed three times with PBS, and then cells were examined by an OLYMPUS FV1200 laser scanning confocal microscope.

### 4.14. FACS/Phosflow

A PE mouse anti-mTOR (pS2448), PE mouse anti-AKT (pS473), and PE mouse IgG1 kappa isotype control were measured by FCAS according to the manufacturer’s instructions. Briefly, cells (2 × 10^5^ cells/mL) were seeded in 12-well plates and cultured overnight. The cells were treated with 50, 150, and 200 μmol/L ADC for 24 h. Medium containing 5‰ DMSO served as the negative control. After treatments, cells were trypsinized and washed with cold stain buffer, and then fixed with cold fixation buffer for 30 min at 4 °C. After washing with staining buffer, cells were permeabilized with perm buffer III for 30 min at 4 °C. Cells were washed and resuspended in staining buffer, and 100 μL cell suspension to each test was continued with antibody staining for 1 h at room temperature. Then cells were washed with staining buffer and analyzed using FCAS, and the results were analyzed using Flowjo software.

### 4.15. Role of Autophagy

To explore the exact role of autophagy in the anticancer action of ADC in NSCLC, the autophagy inhibitor CQ and autophagy activator RAPA were used to treat the cells after ADC treatment. Cells were treated with 200 μM ADC for 72 h, and then were treated with 100 μM CQ and 5 μM RAPA for 4 h, respectively. The proliferation was analyzed by alamarBlue^@^ assay as shown in Section 4.3. To further confirm whether autophagy takes parts in ADC-induced apoptosis, cells were treated with 50, 150, and 200 μM ADC for 72 h, and then cells were treated with 100 μM CQ for 4 h. The apoptosis rate was analyzed by the Annexin V-FITC apoptosis detection kit and FCAS as shown in Section 4.7.

To determine at which stage (before or after ADC treatment) the protective autophagy occurs in NSCLC with ADC treatment, RAPA was used to treat the cells after ADC treatment. Cells were treated with the 5 μM RAPA for 4 h, and then cells were treated with 200 μM antrodin C for 72 h. The proliferation was analyzed by alamarBlue^@^ assay as shown in Section 4.3.

### 4.16. In Vitro ADC Metabolism

Many pharmacologically interesting molecules must be passed over because they are not sufficiently stable. Thus, it is necessary to focus on metabolic stability, which is widely considered one of the most significant challenges of drug discovery. The Phase I metabolism assay was performed by incubation of ADC (final concentrations, 1 μM) with an NADPH regenerating system (final concentration, 1 unit/mL) in SD rat and human liver microsomes (final concentration, 0.5 mg/mL). DMSO stock solution of ADC was prepared at a concentration of 10 mM. Working solution of ADC was prepared by adding methanol at 100 μM, and then by adding phosphate buffer at 10 μM. Liver microsome working solution was prepared by phosphate buffer at a concentration of 0.625 mg/mL. Eighty microliters of liver microsome working solution was added to the 10 μL ADC working solution directly, and 10 μL NADPH regenerating system solution were mixed, and then incubated at 37 °C for 0, 5, 10, 20, 30, and 60 min to initiate the reaction, followed by quenching of the reaction by adding 300 μL cold acetonitrile containing internal standards (100 ng/mL Tolbutamide) to each well. The Blank60 plate was referred to the no-test-compound treatment at 60 min incubation. The NCF60 plate was referred to the treatment of not adding the NADPH regenerating system at 60 min after incubation. All of the plates were centrifuged at 4000 rpm for 20 min, and the supernatants were transferred to new plates for analysis by LC/MS/MS. 

### 4.17. Statistical Analyses

Results are presented as means ± standard deviation (SD). Inter-group comparisons were performed by one-way analysis of variance (ANOVA) and LSD’s test. All of the variables were tested for normal and homogeneous variance using Levene’s test. When necessary, Tamhane’s T2 test was performed. A *p*-value of less than 0.05 or 0.01 was significant and very significant, respectively.

## 5. Conclusions

In summary, we provided in vitro evidence in human lung adenocarcinoma cancer cell lines which suggests that ADC retards cell growth, represses cell migration, disturbs cell cycle progression, and induces apoptotic death. We further demonstrated that protective autophagy could occur simultaneously in lung cancer cells exposed to ADC, and that these changes were partially mediated by the Akt-mTOR pathway. These findings may be helpful in the development of ADC as a chemotherapeutic lead compound for lung cancer, as well as to the rationale for enhancing its anti-lung cancer efficacy through the inhibition of protective autophagy.

## Figures and Tables

**Figure 1 molecules-24-00993-f001:**
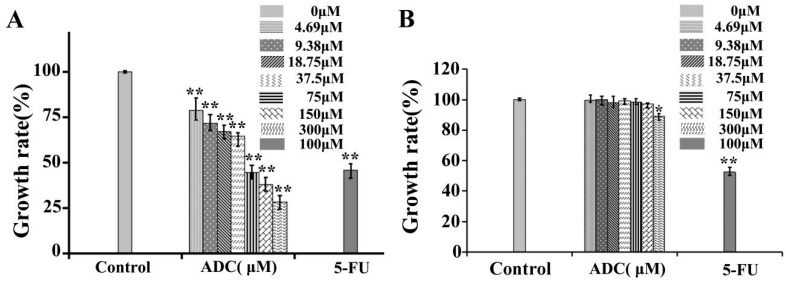
In vitro cell growth–inhibitory activity of ADC. SPCA-1 (**A**) and BEAS-2B (**B**) cell growth inhibition rates are shown after the cells were treated with agents at the indicated concentration for 72 h. The different agents were dissolved and applied in DMSO. 5-FU was used as a positive control * *p* < 0.05, ** *p* < 0.01 vs. control.

**Figure 2 molecules-24-00993-f002:**
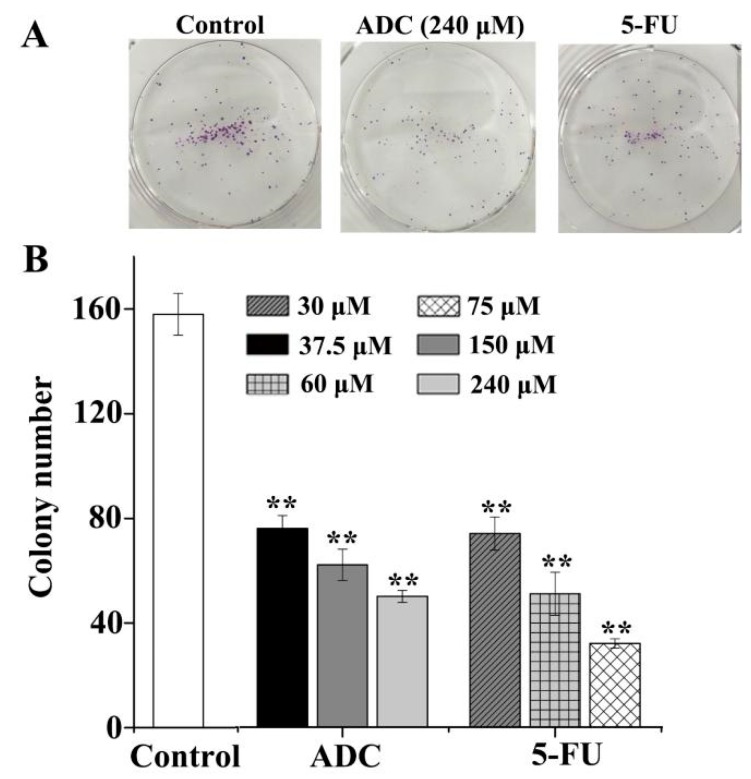
Colony formation assay. (**A**) ADC inhibited colony formation in SPCA-1 cells. After being treated with or without ADC, cells were seeded at 100 cells per plate and allowed to form colonies. After two weeks, the numbers of colonies were counted and recorded. (**B**) Quantification of colony formation. ADC treatment resulted in a significant decrease in colony numbers of cells when compared with untreated cells. 5-FU was used as a positive control ** *p* < 0.01 vs. control.

**Figure 3 molecules-24-00993-f003:**
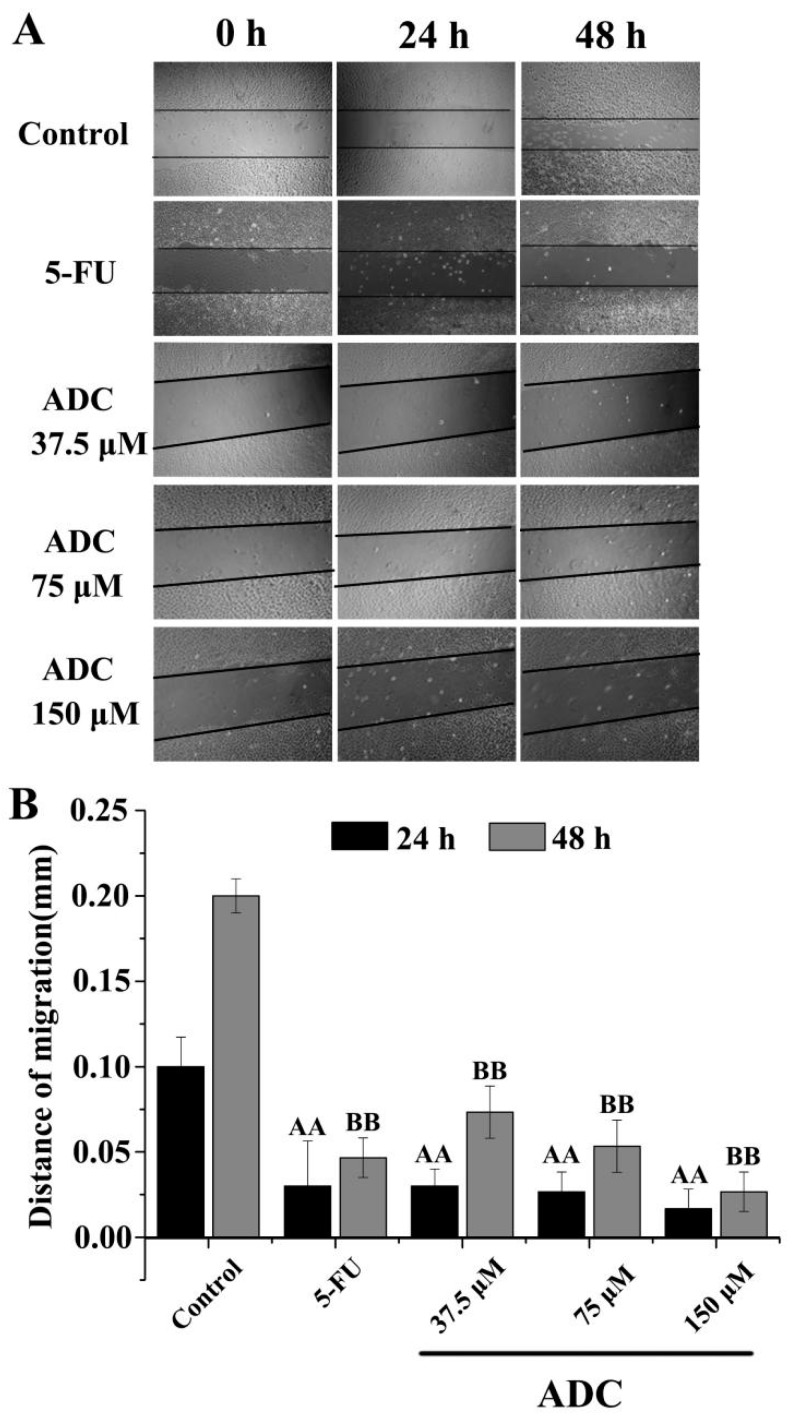
Cell migration assay. A scratch was made on the cell monolayer before treatments (0 h) to serve as a reference for observing cell migration. Photographs were taken at 24 and 48 h time points. ^AA^
*p* < 0.01 vs. control (24 h). ^BB^
*p* < 0.01 vs. control (48 h).

**Figure 4 molecules-24-00993-f004:**
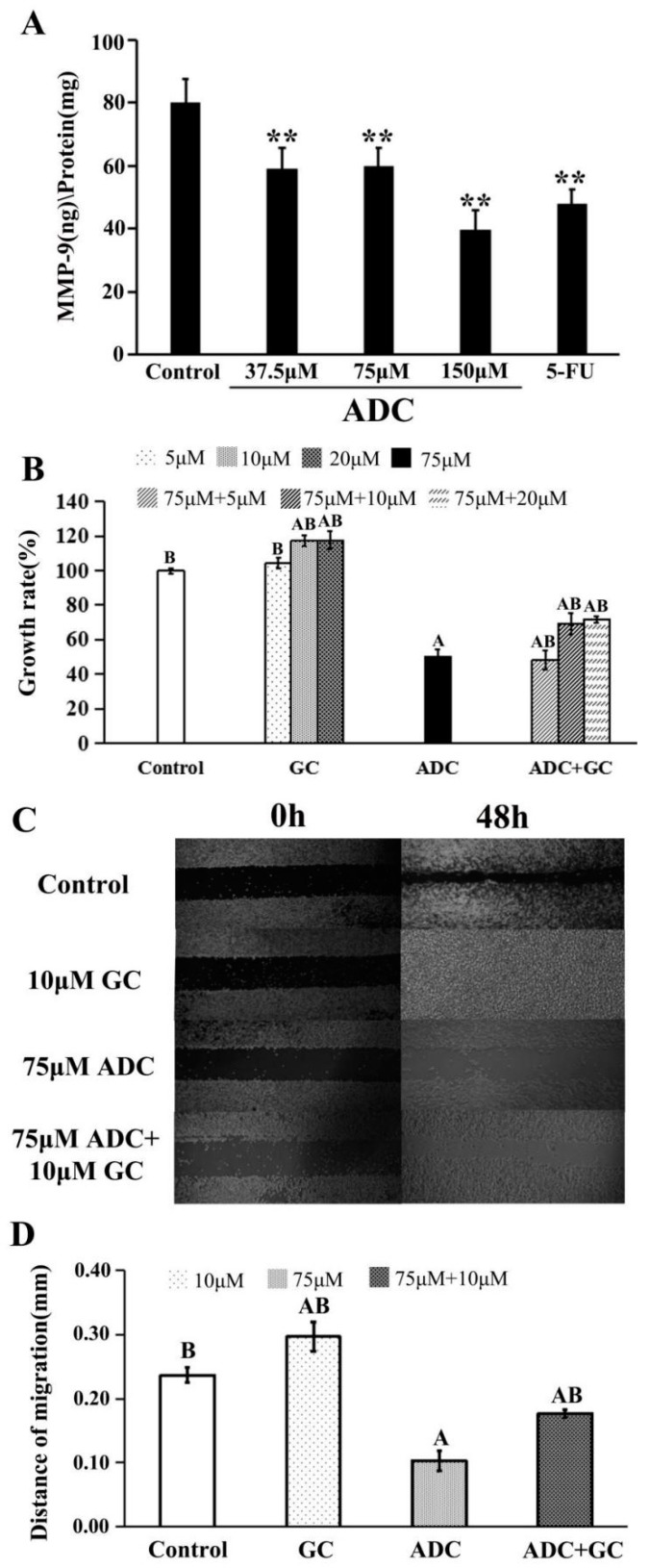
Effects of matrix metalloproteinase (MMP)-9 on the migration and proliferation of SPCA-1. (**A**) Expression of MMP-9 in SPCA-1 was detected by an ELISA kit after being treated with ADC for 72 h, and 5-FU was used as a positive control. ** *p* < 0.01 vs. control; (**B**) effects of MMP-9 activator (GC) on ADC-induced cell growth inhibition were detected by alamarBlue^®^ assay, ^A^
*p* < 0.01 vs. control, ^B^
*p* < 0.01 vs. 75 μM ADC; (**C**) and (**D**) effects of the MMP-9 activator (GC) on ADC-induced cell migration inhibition were detected by microscope, ^A^
*p* < 0.01 vs. control, ^B^
*p* < 0.01 vs. 75 μM ADC.

**Figure 5 molecules-24-00993-f005:**
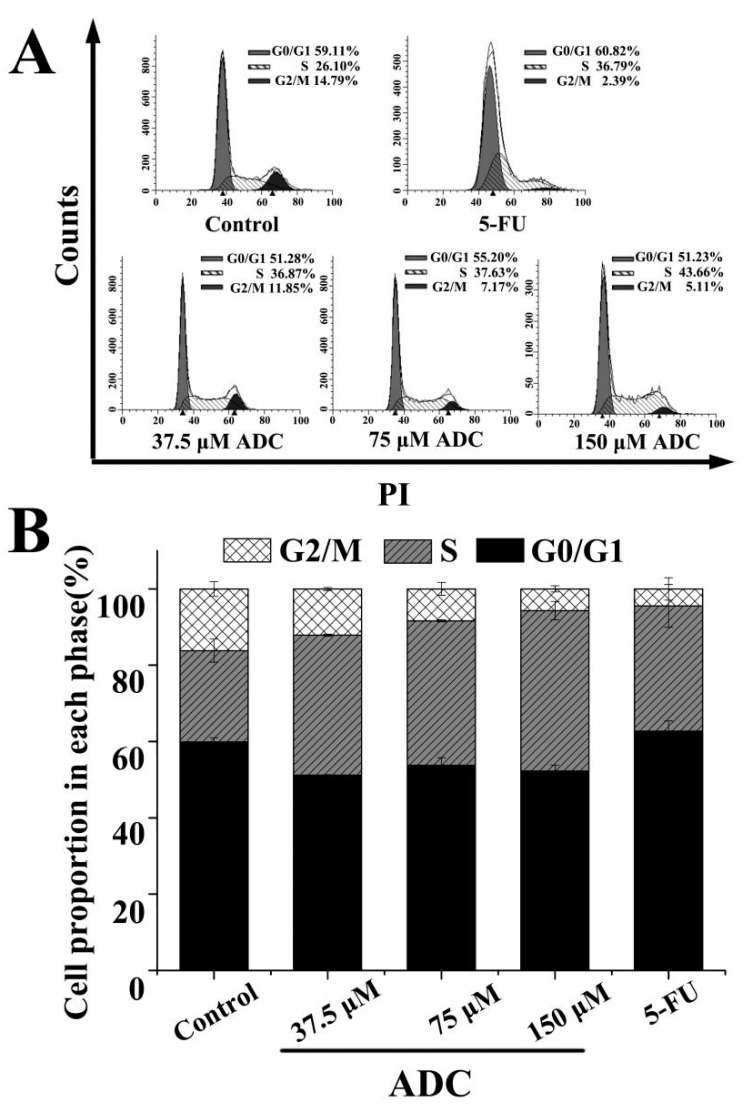
**a** fluorescence-activated cell sorter (FCAS) analysis of the cell cycle. (**A**) SPCA-1 cells were treated using 37.5, 75, and 150 μM ADC for 72 h. 100 μM 5-FU served as a positive control. The distribution of cell cycle, which was stained by PI, was analyzed using FCAS; (**B**) the percentage of cells in each phase of the cell cycles were calculated.

**Figure 6 molecules-24-00993-f006:**
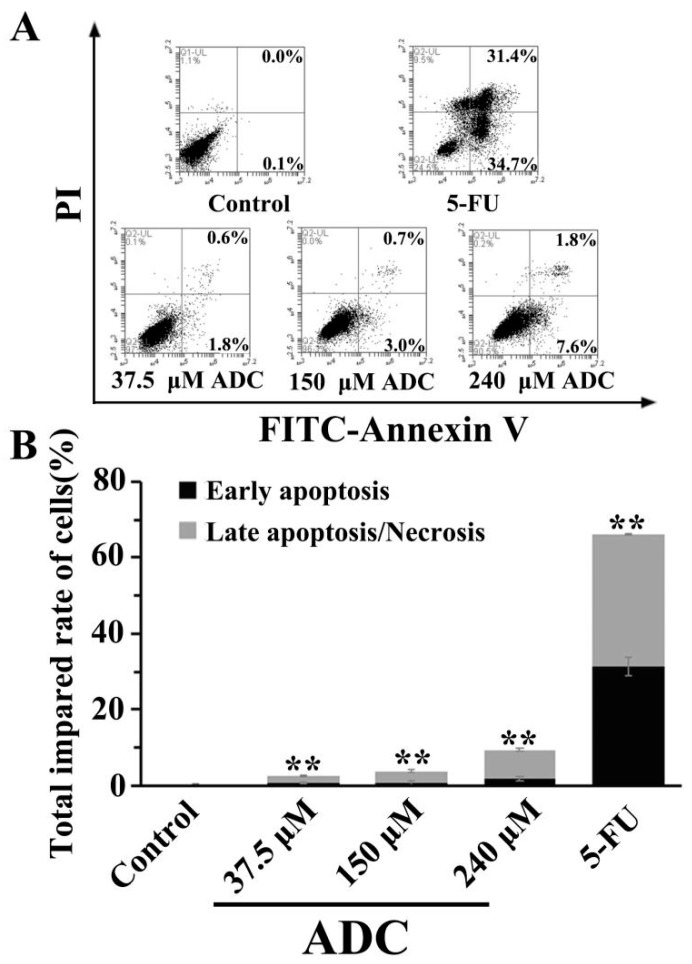
FCAS analysis of apoptosis. (**A**) Cells were incubated with the indicated doses (37.5, 150, and 240 μM) of ADC for 72 h and apoptotic cells were observed with FCAS following annexin V-FITC/PI double staining. (**B**) The total impaired rate of cells in each group was calculated, ** *p* < 0.01 vs. control.

**Figure 7 molecules-24-00993-f007:**
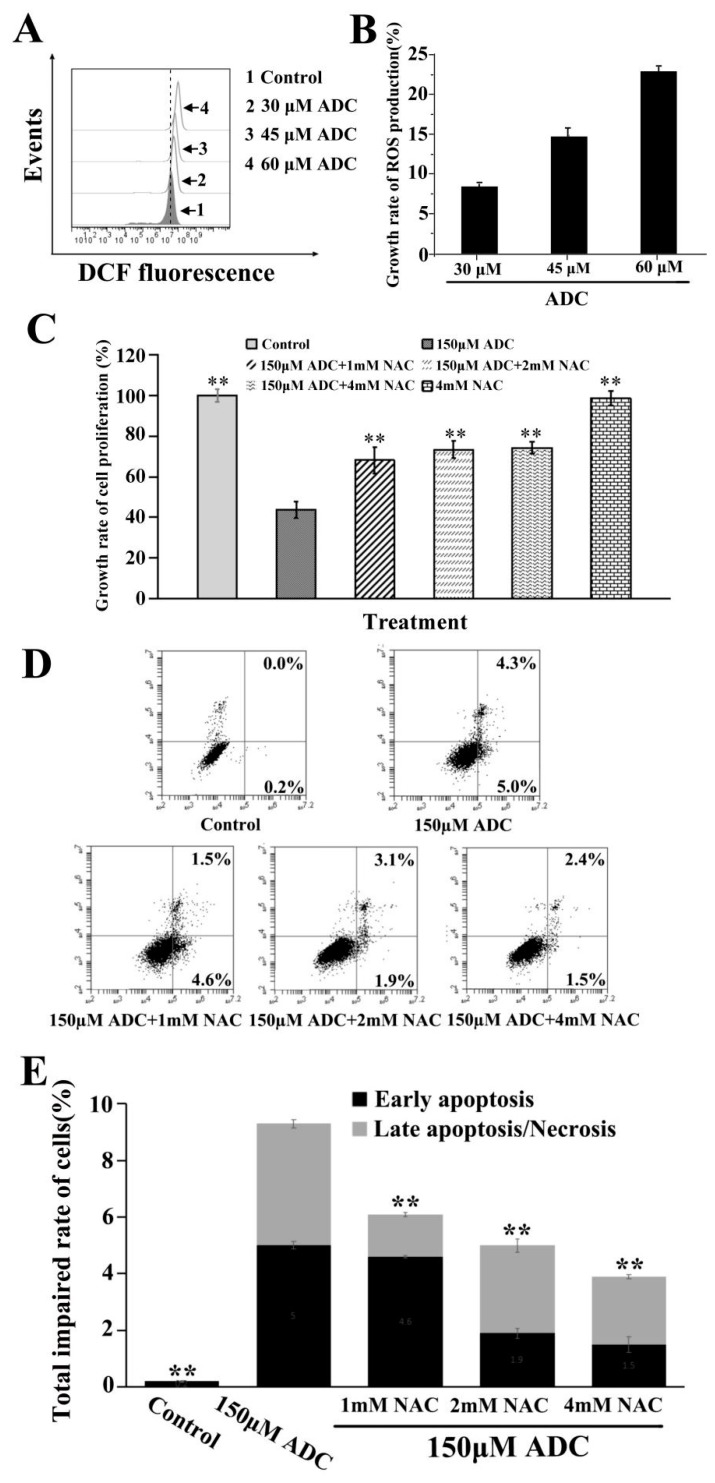
Reactive oxygen species (ROS) detection. (**A**) After being treated with ADC for 24 h, SPCA-1 cells were treated with redox-sensitive fluorescent dye DCFH-DA for 0.5 h, while the ROS generation was determined using FCAS. (**B**) The growth rate of ROS production was calculated. (**C**) Cells were pre-treated with N-acetyl-L-cysteine (NAC) for 2 h prior to exposure to ADC for 72 h, and finally the proliferation rate of cells was detected. ** *p* < 0.01 vs. 150 μM NAC. (**D**) Cells were pre-treated with NAC for 2 h prior exposure to ADC for 72 h, and apoptotic cells were observed with FCAS following annexin V-FITC/PI double staining. (**E**) The total impaired rate of cells in each group was calculated, ** *p* < 0.01 vs. 150 μM ADC.

**Figure 8 molecules-24-00993-f008:**
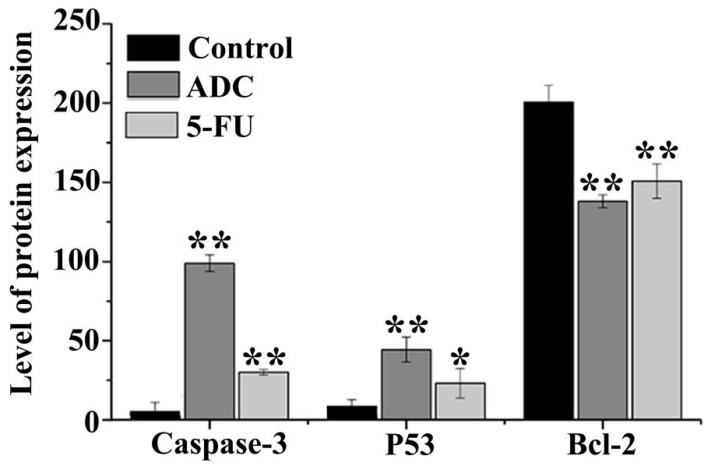
Expression of cleaved caspase-3, Bcl-2, and P53 in cells treated with ADC. Following treatment with 100 μM ADC for 72 h, cleaved caspase-3 and P53 were increased. The expression of Bcl-2 was reduced after ADC treatment for 72 h. * *p* < 0.05, ** *p* < 0.01 vs. control.

**Figure 9 molecules-24-00993-f009:**
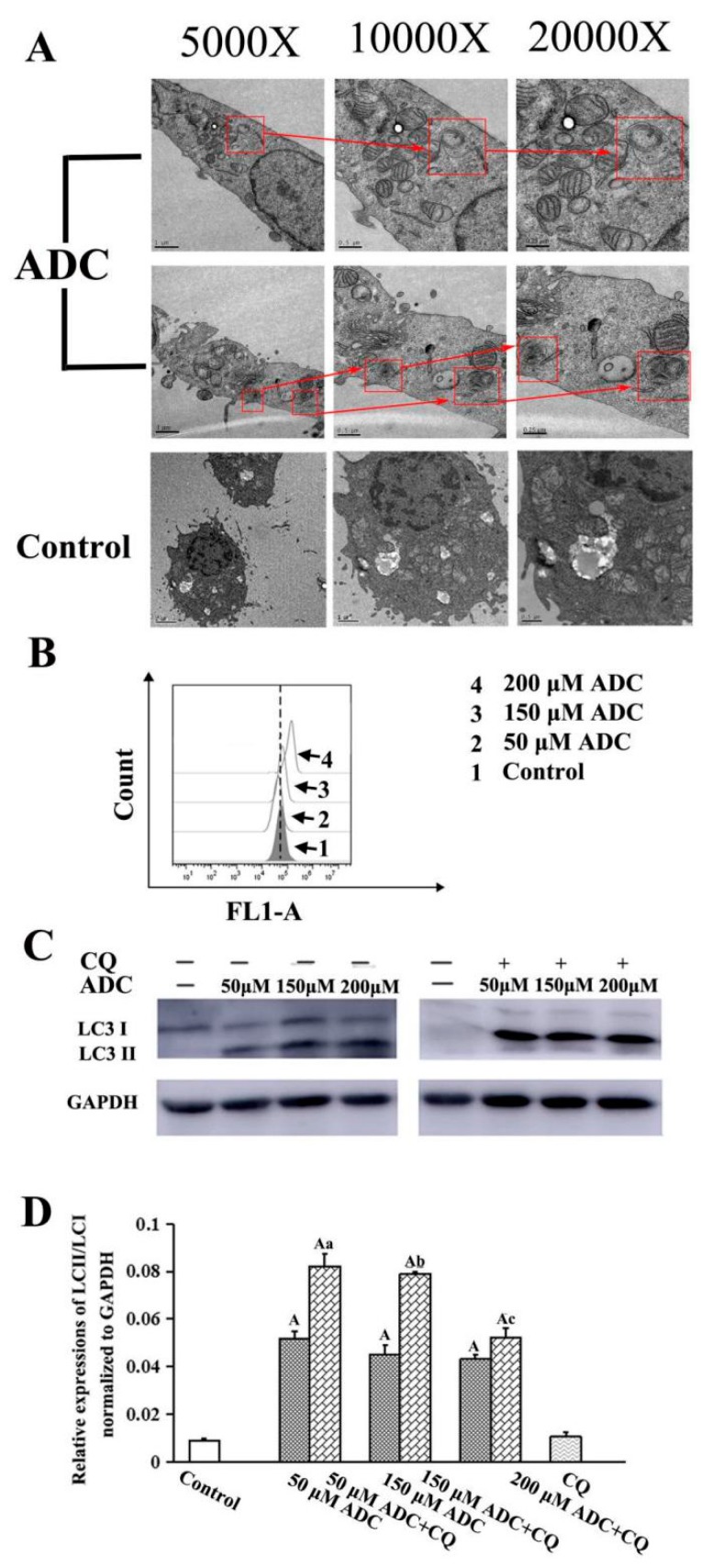
Autophagy induced by ADC in SPCA-1 cells. (**A**) Cytoplasm vacuolization enclosed in a double membrane in SPCA-1 cells was found using TEM. The arrow and rectangular frame indicate autophagic vacuoles; (**B**) conversion of LC3-I to LC3-II in ADC-treated SPCA-1 was found using FCAS; (**C**) effects of autophagy inhibitors (CQ) on ADC-induced cell growth inhibition was detected using Western blotting; (**D**) bar graphs demonstrated the expression of LC-II/LC-I. Data are the mean ± SEM of three independent experiments. ^A^
*p* < 0.01 vs. control; ^a^
*p* < 0.01 vs. 50 μM ADC; ^b^
*p* < 0.01 vs. 150 μM ADC; ^c^
*p* < 0.01 vs. 200 μM ADC.

**Figure 10 molecules-24-00993-f010:**
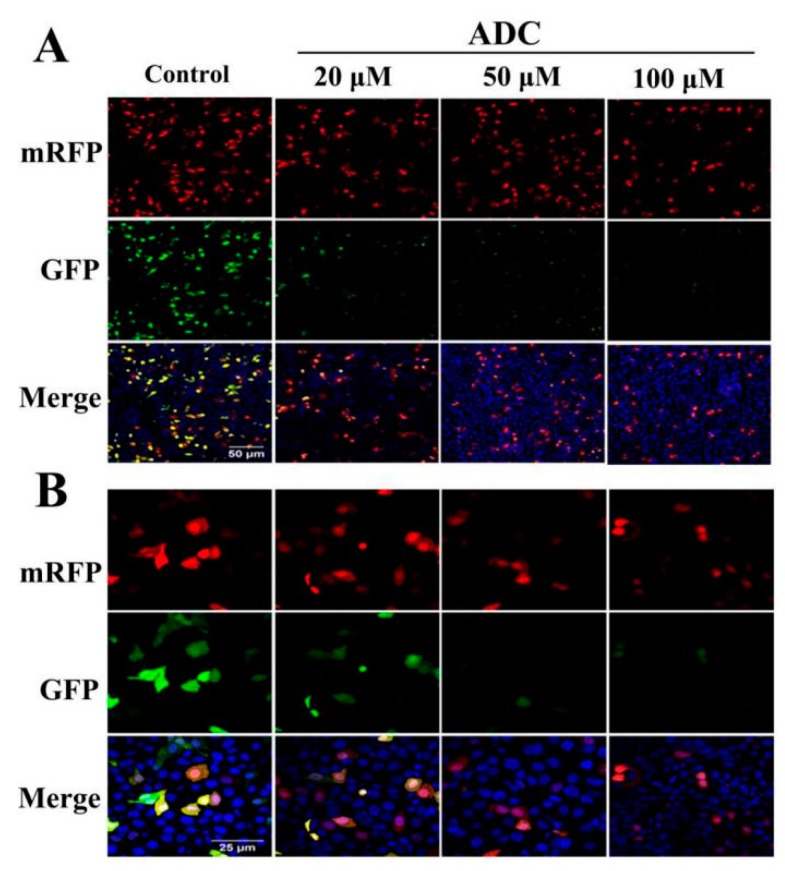
ADC induced autophagy flux. The living cell image was found using a laser scanning confocal microscope. The images were then magnified 30 times (**A**) and then 60 times (**B**).

**Figure 11 molecules-24-00993-f011:**
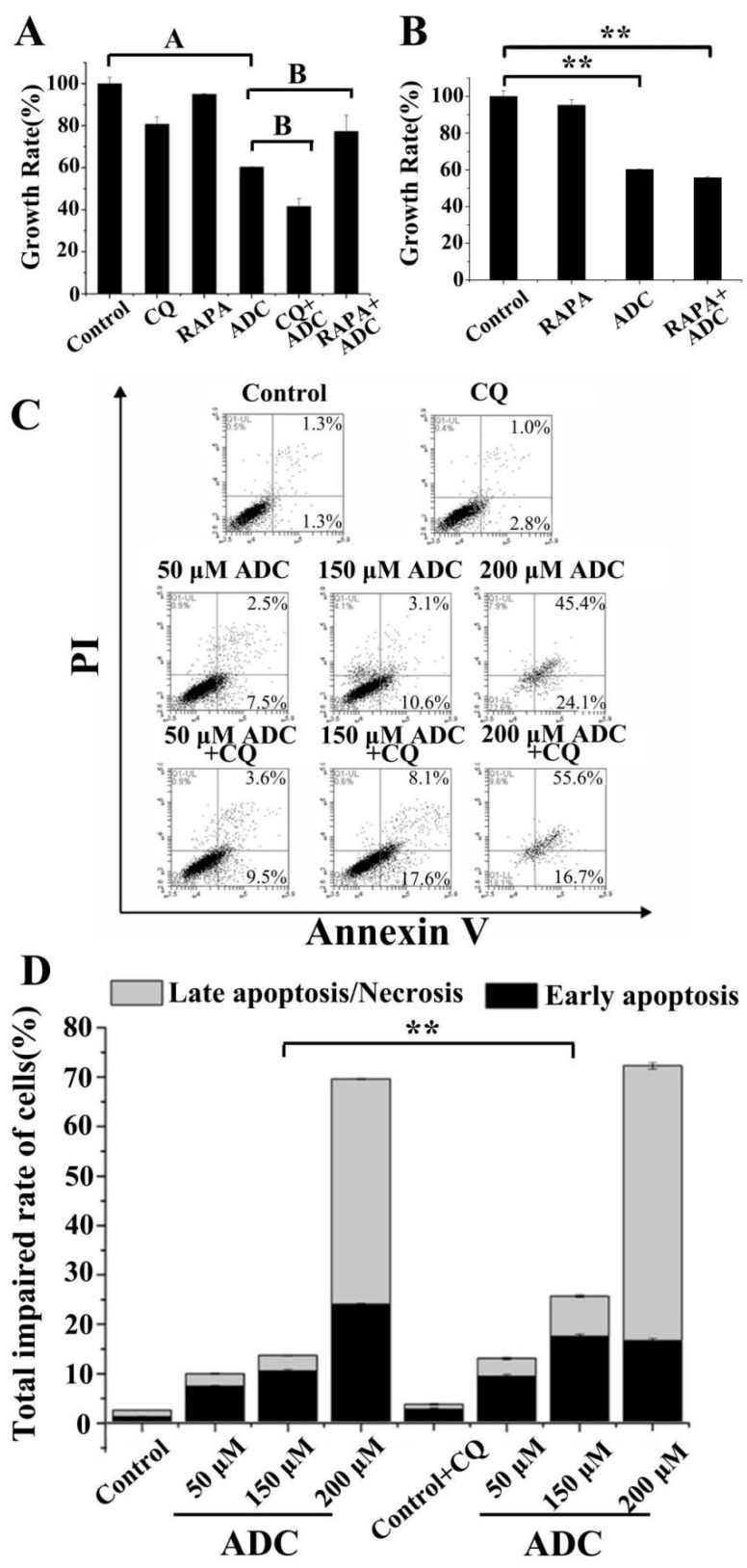
ADC induced protective autophagy. (**A**) Autophagy inhibition was able to protect cells from death. Cells which were pre-treated with ADC were treated with RAPA, and the proliferation rate of cells’ death was detected. ^A^
*p* < 0.01 vs. control; ^B^
*p* < 0.01 vs. ADC. (**B**) Cells were pre-treated with RAPA for 3 h, then treated with ADC for 72 h, and finally the proliferation rate of cells was detected. ** *p* < 0.01 vs. control. (**C**) Cells were incubated with 50, 150, and 200 μM ADC for 72 h, and then treated with CQ. The early and late/necrosis apoptotic cells were observed with FCAS after annexin V-FITC/PI double staining. (**D**) The percentage of apoptosis in SPCA-1 was calculated. ** *p* < 0.01, 150 μM ADC vs. 150 μM ADC + CQ (total impaired rate of cells).

**Figure 12 molecules-24-00993-f012:**
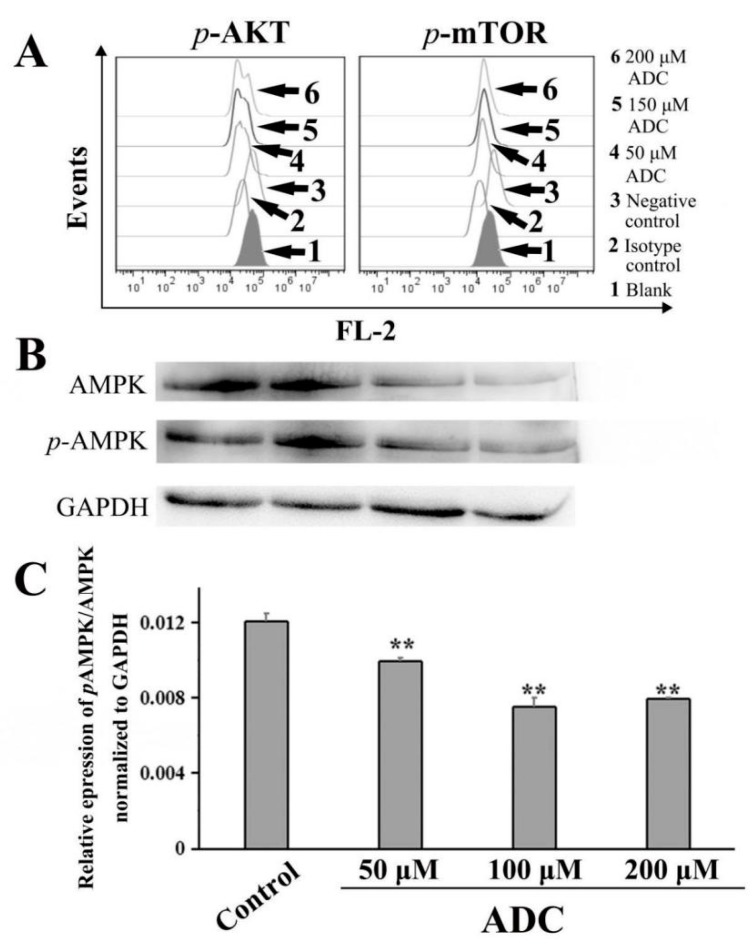
ADC-induced protective autophagy is attributed to the AMPK inhibition-independent blockade of the Akt/mTOR pathway. (**A**) ADC downregulated the AKT-mTOR pathway. Following treatment by ADC, cells were detected by FCAS. (**B**) ADC downregulated the AMPK pathway. Following treatment by ADC, cells underwent lysis and were detected by Western blotting. (**C**) Bar graphs demonstrate the relative expression of pAMPK/AMPK. Data are the mean ± SEM of the three independent experiments. ** *p* < 0.01 vs. control.

**Figure 13 molecules-24-00993-f013:**
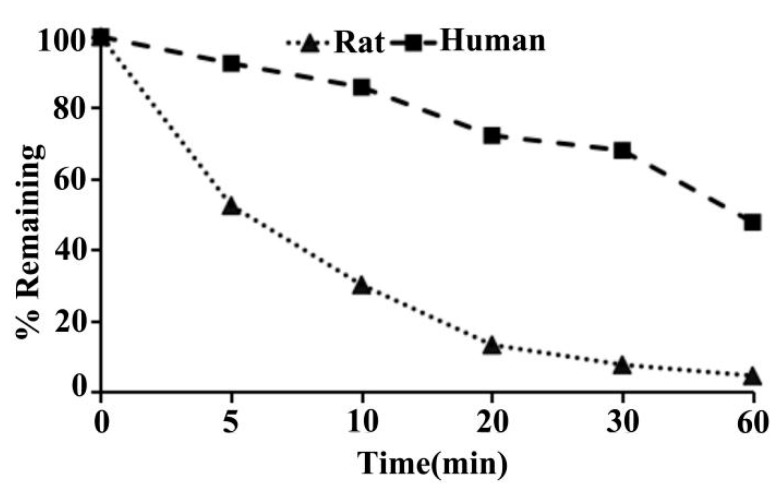
Percentage remaining versus time profile of ADC in SD rat and human liver microsomes.

**Table 1 molecules-24-00993-t001:** Metabolic stability of ADC in SD rat and human liver microsomes following incubation in the presence of NADPH.

Species	T_1/2_ (min)	CL_int(mic)_ (μL/min/mg protein)	CL_int(liver)_ (mL/min/kg)
SD rat	7.5	185.8	334.4
Human	54.1	25.6	23.0

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
