# Peer review of "Antrodin C, an NADPH Dependent Metabolism, Encourages Crosstalk between Autophagy and Apoptosis in Lung Carcinoma Cells by Use of an AMPK Inhibition-Independent Blockade of the Akt/mTOR Pathway"

_molecules, 2019, doi:10.3390/molecules24050993_

Reviewer 1 Report

The authors has substantial data to describe their hypothesis. However, data analysis, presentation and manuscript preparation was very poor. For example, statistical analysis were missing in Figure 2 and 3 or wrongly denoted in Figure 4 & 6. The concentration ranges were vary in each experiments. Overall, the language of the manuscript and structure of the article must be improved prior to re-submission.  

Author Response

The authors has substantial data to describe their hypothesis. However, data analysis, presentation and manuscript preparation was very poor.

1. Statistical analysis were missing in Figure 2 and 3 or wrongly denoted in Figure 4 & 6.

Response: Thank for your suggestion. The errors in figures had been corrected. The revised content has been marked in red in the revised manuscript.

2. The concentration ranges were vary in each experiments

Reponse: Thank for your suggestion. In the preliminary experiment, 200 µM ADC caused cells to float, and so cell migration experiments couldn’t be carried out. In the preliminary experiment of cell cycle, we also found the cell cycle could not be detected by the FACS when the SPCA-1 cell was treated with 200 uM ADC. In preliminary experiment, the effect of gradient concentration of ADC (from 10 to 200uM) on ROS release from SPCA-1 were analysis. We only found the rate of ROS release was positive correlation with ADC treatment, which ranged from 30 μM to 60 μM. So in the formal experiment, we only explored the effect of 30, 45 and 60 μM ADC on ROS release.

3. Overall, the language of the manuscript and structure of the article must be improved prior to re-submission

Response: We have consulted with an English language editing company to improve the manuscript. PDF proofs have also been uploaded as a supplemental document.

Reviewer 2 Report

The authors have addressed all my concerns.

Author Response

Thank for your suggestion.We have consulted with an English language editing company to improve the manuscript. PDF proofs have also been uploaded as a supplemental document.

Reviewer 3 Report

NA

Author Response

Thank for your suggestion.We have consulted with an English language editing company to improve the manuscript. PDF proofs have also been uploaded as a supplemental document.

Round  2

Reviewer 1 Report

This authors have provided extra attention to revise the manuscript. Here, would suggest for possible publication in Molecules

This manuscript is a resubmission of an earlier submission. The following is a list of the peer review reports and author responses from that submission.

Round  1

Reviewer 1 Report

In this manuscript the authors trying to compile the possible molecular mechanisms by which ADC regulates in lung cancer (SPCA-1) cell line. However, to give a complete picture of the anti-cancer effects of ADC, the authors are advised to address the following points before publication.

Comments:

The concentrations of ADC used in this study seems very high when compared to previous studies by SY Wang’s Lab. They found that above 20 µM of ADC exhibited cytotoxicity to MCF-7 breast cancer and HUVECs. Therefore, the authors advised to examine the cytotoxicity of ADC on normal lung cells.

It is not clearly explained why different concentrations used for different experiments?

In vitro migration: As shown in figure 3, ADC inhibits migration of SPCA-1 cells at 37.5 µM. whereas, the same concentration reduced cell growth to below 70% (Figure 1). Therefore, it can be hypothesis that the reduced of migration by ADC may due to the growth inhibition. To further delineate, the authors must include the effects of ADC on migration regulatory proteins such as matrixmetalloprotenases and plasminogens.

Apoptosis: Figure 5A, the positive control increased early and late apoptosis to 34.7% and 31.4%, respectively. However, in the histogram data (panel B) expressed ~3.5% and ~5%, respectively. Likewise, the percentage of apoptosis represented in histogram was totally impaired to the FACS data. In addition, line 143, 240 µM of ADC increased apoptosis to 943.51% or 9.43%?

ROS: It has been well demonstrated that low levels of ROS increase tumor cell survival, whereas high concentration of ROS induced apoptosis through oxidative stress. Here the authors claim that ADC-induced apoptosis was triggered by ROS. To further confirm this issue, the authors are advised to perform apoptosis assay under pharmacological inhibitor of ROS. Also, why the authors used intermediate concentration of ADC to determine ROS production?

In addition, previous reports from SY Wang’s lab exhibiting that treatment with ADC prevents apoptosis via inhibition of ROS generation in human endothelial cells. The authors must discuss this briefly.

Figure 6B, its is confusing that why the growth rate was significantly increased in parallel to ROS generation?

Figure 10. Statistical representation in figures were confusing.

The authors are advised to use the terms “control and 5-FU” instead of “negative control and positive control”

Line 466, the manufacturer’s details must be included.

There are plenty of typographical errors. The authors must provide extra attention to correct it.

Author Response

Reviewer 1 Comments and Suggestions for Authors In this manuscript the authors trying to compile the possible molecular mechanisms by which ADC regulates in lung cancer (SPCA-1) cell line. However, to give a complete picture of the anti-cancer effects of ADC, the authors are advised to address the following points before publication. Comments: 1.The concentrations of ADC used in this study seems very high when compared to previous studies by SY Wang’s Lab. They found that above 20 µM of ADC exhibited cytotoxicity to MCF-7 breast cancer and HUVECs. Therefore, the authors advised to examine the cytotoxicity of ADC on normal lung cells. Response: Thank for your suggestion. The tumor cytotoxicity without damage on normal lung epithelial cells(BEAS-2B) were confirmed. And supplementary data were provided and marked in red in the reviesed manuscript. (Fig.1B and Line 90-96) 2.It is not clearly explained why different concentrations used for different experiments? Reponse: Thank for your suggestion. In the preliminary experiment, 200 µM ADC caused cells to float, and so cell migration experiments couldn’t be carried out. In the preliminary experiment of cell cycle, we also found the cell cycle could not be detected by the FACS when the SPCA-1 cell was treated with 200 uM ADC. In preliminary experiment, the effect of gradient concentration of ADC (from 10 to 200uM) on ROS release from SPCA-1 were analysis. We only found the rate of ROS release was positive correlation with ADC treatment, which ranged from 30 μM to 60 μM. So in the formal experiment, we only explored the effect of 30, 45 and 60 μM ADC on ROS release. 3.In vitro migration: As shown in figure 3, ADC inhibits migration of SPCA-1 cells at 37.5 µM. whereas, the same concentration reduced cell growth to below 70% (Figure 1). Therefore, it can be hypothesis that the reduced of migration by ADC may due to the growth inhibition. To further delineate, the authors must include the effects of ADC on migration regulatory proteins such as matrix metalloprotenases and plasminogens. Response:Thank for your suggestion. And supplementary data were provided and marked in red in the revised manuscript. (Line124-145 and 491-518) 4.Apoptosis: Figure 5A, the positive control increased early and late apoptosis to 34.7% and 31.4%, respectively. However, in the histogram data (panel B) expressed ~3.5% and ~5%, respectively. Likewise, the percentage of apoptosis represented in histogram was totally impaired to the FACS data. In addition, line 143, 240 µM of ADC increased apoptosis to 943.51% or 9.43%? Response: Sorry for my mistakes in Figure 5B, and I have corrected these mistakes in revived manuscript. Sorry for miscalculating the apoptotic growth rate.The average of total impaired rate of cells in control and 240uM ADC groups were 0.1% and 9.4%, and the growth rate of 240uM ADC group was 93%. The average growth rate of 240 uM group was calculated as follwoing: (9.4%-0.1%)/0.1%=93%. The revised content has been marked in red in the revised manuscript. (Line173-174 and Figure 6) 5.ROS has been well demonstrated that low levels of ROS increase tumor cell survival, whereas high concentration of ROS induced apoptosis through oxidative stress. Here the authors claim that ADC-induced apoptosis was triggered by ROS. To further confirm this issue, the authors are advised to perform apoptosis assay under pharmacological inhibitor of ROS. Also, why the authors used intermediate concentration of ADC to determine ROS production? Response: Thank for your suggestion. The N-acetyl-L-cysteine (NAC) is known as ROS scavenger. In this study, the NAC was used to explore the role of ROS on proliferation and apoptosis of SPCA-1 induced by ADC.The relevant analysis has been added and marked in red in the revised manuscript.(Figure 7 and Line 189-194, 198-203 and 547-556)  In preliminary experiment, the effect of gradient concentration of ADC (from 10 to 200uM) on ROS release from SPCA-1 were analysis. We only found the rate of ROS release was positive correlation with ADC treatment, which ranged from 30 μM to 60 μM. So in the formal experiment, we only explored the effect of 30, 45 and 60 μM ADC on ROS release. 6.In addition, previous reports from SY Wang’s lab exhibiting that treatment with ADC prevents apoptosis via inhibition of ROS generation in human endothelial cells. The authors must discuss this briefly. Response: Thank for your suggestion, and the relevant analysis has been added and marked in red in the revised manuscript.(Line from 362 to 377) 7.Figure 6B, its is confusing that why the growth rate was significantly increased in parallel to ROS generation? Response: I'm sorry to confuse you because we don't have a clear axis title in Figure 7B. The ordinate values of Figure 7B do not represent the growth rate of cell proliferation, but the growth rate of ROS release. Now I have revised the axis title in Figure 7B. 8.Figure 10. Statistical representation in figures were confusing. Response: Thank for your suggestion. I have revised the statistical representation in Figure11. In Figure 11A, the autophagy inhibitor CQ and activator RAPA were used to explored the exact role of autophagy on cell proliferation of SPCA-1 induced by ADC. The statistical data showed a significant decrease(38.49%) in ADC treatment group, as compared with the control group. And we also found a significant decrease (18.74%) or increase (17.08%) was in proliferation rate of SPCA-1, of which autophagy was inhibited by CQ or activated by RAPA after ADC treatmeat, in contrast to treatemet of ADC alone. These results indicated that ADC-induced autphagy is protective. In Figure 11B, the RAPA was further used to explore the extract stage of protective autophagy, and the result showed pretreatment with RAPA didn’t restored proliferation inhibition of SPCA-1 induced by ADC. Combined the results of RAPA in Figure 11A and Figure 11B suggested the survival effect occurred only when cells were under stress. In Figure 11C and 11D, the autophagy inhibitor CQ was used to explore the exact role of autophagy on SPCA-1 apoptosis induced by ADC. The statistical data showed apototic cells were increased 1-fold by combination treatment with 150 µM ADC and CQ, compared with treatment of 150 µM ADC alone. These reuslts indicated ADC induced protective autophagy in SPCA-1 cells. 9.The authors are advised to use the terms “control and 5-FU” instead of “negative control and positive control” Response: Thank for your suggestion, “negative control and positive control” has been instead by “control and 5-FU” in revised manuscript. 10.Line 466, the manufacturer’s details must be included. Response: Thank for your suggestion, the relevant analysis has been added and marked in red in the revised manuscript.(Line 566-576) 11.There are plenty of typographical errors. The authors must provide extra attention to correct it. Response: We have consulted with an English language editing company to improve the manuscript. PDF proofs have also been uploaded as a supplemental document.

Reviewer 2 Report

This is an interesting paper these are few things authors can consider addressing

The data generated was from one cell line, it will be ideal to show some key results in few cell lines. This will increase robustness of these findings.

The authors showed Antrodin C increases ROS, it would be interesting to demonstrate addition of antioxidants abrogates the changes seen with Antrodin C. Which will confirm Antrodin C mechanisms of action is regulated by ROS.

Author Response

Comments and Suggestions for Authors

This is an interesting paper these are few things authors can consider addressing

1. The data generated was from one cell line, it will be ideal to show some key results in few cell lines. This will increase robustness of these findings.

Response: Thank for your suggestion. In our research, the experimental data about anti-lung cancer activity of ADC on A549 and HCC827 have been sorted out and written into two articles, one entitled “Increased Inhibition Effect of Antrodin C from Stout the Camphor Medicinal Mushroom, Taiwanofungus camphoratus (Agaricomycetes), on A549 through Crosstalk between Apoptosis and Autophagy”, and the other one entitled ”Anti-tumor Effect on HCC827 Lung Adenocarcinoma Cell by Antrodin C from Spent Broth from Submerged Cultures of Taiwanofungus camphoratus”. This first one have been accpeted by International Journal of Medicinal Mushroom, and will be published on 2019. The other one will be submittet to molecules. Now some data were chosen from these two manuscripts to show the anti-tumor activity on additional lung cancer cell lines including A549 and HCC827.

As shown in Fig.1, the clone formation of A549 cells was significantly inhibited when being treated with 12.5, 50 and 80 μg/mL antrodin C, and the adherence rates of reducing were in a concentration dependent manner. As shown in Fig.2, the results indicated cells migrated more slowly to close the scratched wounds after the treatment of 50 and 80 μg/mL antrodin C for 48 h(P<0.01).< span=""> As shown in Fig.3, antrodin C inhibited the growth of A549 cells in a concentration and time-dependent manner. 

As shown in Fig.4, the proliferation of HCC827 cells could be inhibited by ADC treatment in a dose-dependent manner. The inhibition rate of ADC at the concentration of 50 μM on HCC827 was 54.17%, and its IC50 was 324.8 μM. It was found ADC could induce HCC827 cells arrest in S-phase in a dose-dependent manner. As shown in Fig.5, after treatment with gradient concentration of ADC (from 50 to 200 μM), percentage of cell cycle at S phase rised from 14.1% to 33.2%. With the increase of ADC concentration, the early apoptosis rate increased from 4.0% to 9.3%, and the total apoptosis rate increased from 9.3% to 20.8%(Fig.6). Compared with the negative control, the total apoptosis rate increased by 15.1 times(Fig.6).

2. The authors showed Antrodin C increases ROS, it would be interesting to demonstrate addition of antioxidants abrogates the changes seen with Antrodin C. Which will confirm Antrodin C mechanisms of action is regulated by ROS.

Response: Thank for your suggestion. The relevant analysis has been added and marked in red in the revised manuscript.(Figure 7 and Line 189-194, 198-203 and 547-556) 

Reviewer 3 Report

In the manuscript entitled Antrodin C, NADPH dependent metabolism, induces Crosstalk between apoptosis and autophagy in human lung cancer cells through AMPK inhibition-independent blockade of Akt/mTOR pathway” the authors have evaluated the anti-lung cancer activity of antrodin C (ADC), a maleimide derivative isolated from submerged mycelium of Taiwanofungus camphoratus. Results showed that ADC potently inhibited viability of SPCA-1cells, induced ROS-triggered apoptosis and G2 cell cycle arrest through caspase and P53 dependent pathway. Interestingly, our results also showed that ADC treatment activated autophagy in SPCA-1 cells, as evidenced by the accumulation of autophagosomes et al. Blockage of autophagy augmented ADC induced cell death, and activation of autophagy restored cell death, suggesting that autophagy played a protective role in ADC treated cells. Meanwhile, ADC treatment suppressed Akt-mTOR pathway and AMPK pathway. Additionally, the combination of ADC with autophagy inhibitor significantly increased the cells death of SPCA-1 cells. In vitro phase I metabolic stability assay showed that ADC was highly metabolized in SD rat liver microsomes, and moderately metabolized in human liver microsomes, which giving a reference in predicting clinical pharmacokinetics and toxicity studies.

Specific Comments:

This is technically well performed study but the authors need to address several missing links before it can be even considered for publication. Specific points that the authors need to address are as follows:

The molecular mechanism(s) by which ADC exhibits its anticancer effects are not clear? For  example, whether deletion of Akt/mTOR by siRNA abrogates the observed anticancer effects of ADC should be analyzed?

Most of the experiments have been done in one lung cancer cell line. Additional lung cancer cell lines should be used to validate the key findings of this study.

The effect of ADC should      also be analyzed on normal lung epithelial cells to rule out potential cytotoxicity. Also, acute toxicity studies should be performed to establish the safety of the compound.

Also, the anti-invasive/anti-migratory effects of ADC should be analyzed.

A limited in vivo study in appropriate xenograft/orthotopic mouse model will greatly increase the impact of the reported in vitro findings.

Several typographical errors were noted throughout the manuscript and also even in the abstract.

Author Response

Specific comments:

This is technically well performed study but the authors need to address several missing links before it can be even considered for publication. Specific points that the authors need to address are as follows:

1. The molecular mechanism(s) by which ADC exhibits its anticancer effects are not clear? For example, whether deletion of Akt/mTOR by siRNA abrogates the observed anticancer effects of ADC should be analyzed?

Reponse: Thank for your suggestion. Rapamycin(RAPA) is not only a autophagy activators, but also a specific mTOR inhibitor with IC50 of ~0.1 nM HEK293 cells. In this study, we found a significant increase (17.08%) was in growth rate of SPCA-1, which treated by RAPA after ADC treatment, in contrast to treatment of ADC alone. This result showed that mTOR inhibition could attenuate the inhibition effect of ADC on SPCA-1 proliferation, and so indicated that Akt/mTOR signaling pathway played an important role on anticancer effect of ADC.(Figure 11A in revised manuscript)

2. Most of the experiments have been done in one lung cancer cell line. Additional lung cancer cell lines should be used to validate the key findings of this study.

Response: Thank for your suggestion. In our research, the experimental data about anti-lung cancer activity of ADC on A549 and HCC827 have been sorted out and written into two articles, one entitled “Increased Inhibition Effect of Antrodin C from Stout the Camphor Medicinal Mushroom, Taiwanofungus camphoratus (Agaricomycetes), on A549 through Crosstalk between Apoptosis and Autophagy”, and the other one entitled ”Anti-tumor Effect on HCC827 Lung Adenocarcinoma Cell by Antrodin C from Spent Broth from Submerged Cultures of Taiwanofungus camphoratus”. This first one have been accepted by International Journal of Medicinal Mushroom, and will be published on 2019. The other one will be submitted to molecules. Now some data were chosen from these two manuscripts to show the anti-tumor activity on additional lung cancer cell lines including A549 and HCC827.

As shown in Fig.1, the clone formation of A549 cells was significantly inhibited when being treated with 12.5, 50 and 80 μg/mL antrodin C, and the adherence rates of reducing were in a concentration dependent manner. As shown in Fig.2, the results indicated cells migrated more slowly to close the scratched wounds after the treatment of 50 and 80 μg/mL antrodin C for 48 h(P<0.01). As shown in Fig.3, antrodin C inhibited the growth of A549 cells in a concentration and time-dependent manner. As shown in Fig.4, the proliferation of HCC827 cells could be inhibited by ADC treatment in a dose-dependent manner. The inhibition rate of ADC at the concentration of 50 μM on HCC827 was 54.17%, and its IC50 was 324.8 μM. It was found ADC could induce HCC827 cells arrest in S-phase in a dose-dependent manner. As shown in Fig.5, after treatment with gradient concentration of ADC (from 50 to 200 μM), percentage of cell cycle at S phase rised from 14.1% to 33.2%. With the increase of ADC concentration, the early apoptosis rate increased from 4.0% to 9.3%, and the total apoptosis rate increased from 9.3% to 20.8%(Fig.6). Compared with the negative control, the total apoptosis rate increased by 15.1 times(Fig.6).

3. The effect of ADC should also be analyzed on normal lung epithelial cells to rule out potential cytotoxicity. Also, acute toxicity studies should be performed to establish the safety of the compound.

Response: Thank for your suggestion. The tumor cytotoxicity without damage on normal lung epithelial cells(BEAS-2B) were confirmed. And supplementary data were provided and marked in red in the revised manuscript.

Because of the large amount of compound need in animal experiments, the amount of ADC isolated is not enough to carry out animal experiment, so we choose the mycelium of Taiwanofungus camphoratus to carry out animal experiments. According to the maximum tolerance method, acute toxicity test of the mycelium of Taiwanofungus camphoratus on mice was carried out, and the designed dose was 15000 mg/kg. After 5 days of animal quarantine, the animals fasted for 16 hours and drank water freely before the experiment. The dosage of the sample was 15000mg/kg, the solution was prepared at 750mg/mL with sterile water and gavaged according to 20mL/kg. After 14 days of continuous observation, the poisoning manifestations, death and weight were recorded respectively. After the mice were given the sample by gavage, it was found that the mice in each group had normal diet and activities, grew well, did not show any poisoning and died(Table 1). And the result also showed the sample had no significant effect on weight gain in mice(Table 1). The acute oral maximum tolerated dose (MTD) of the mycelium of Taiwanofungus camphoratus in female and male mice was more than 15000 mg/kg. According to the classification standard of acute toxicity dose(Table.2), the sample belonged to non-toxic grade.

Table 1 Effects of ethanol extract from mycelium of Taiwanofungus camphoratus on acute toxicity of orally treated mice()

Gender

Number of animals

Initial weight (g)

Final weight (g)

Dose (mg/kg·bw)

Number of death

MTD

[mg/(kg·bw)]

Female

10

20.3±0.5

27.8±2.1

15 000

0

>15 000

Male

10

20.9±0.7

33.3±2.0

15 000

0

>15 000

Table 2 Acute toxic dose of scale

Level

LD50 (mg/kg)

Extremely toxic

﹤1

Highly toxic

1–50

Moderately toxic

51–500

Low toxic

501–5 000

Actually nontoxic

5001–15 000

Nontoxic

﹥15 000

4. Also, the anti-invasive/anti-migratory effects of ADC should be analyzed.

Response: Thank for your suggestion. The relevant analysis has been added and marked in red in the revised manuscript. (Line124-145 and 491-518)

5. A limited in vivo study in appropriate xenograft/orthotopic mouse model will greatly increase the impact of the reported in vitro findings.

Response: Thank for your suggestion. In preliminary study, the elimination effect of the mycelium of Taiwanofungus camphoratus on lung adenocarcinoma A549 cells tumor-burdened mice tumor growth was detected. C57BL/6 mice were subcutaneously injected with the tumor cells of A549 consisting of 2×107 cells/L (0.2 mL in total). After 14days of inoculation with A549, the mice were intraperitoneally injected with the mycelium (12.5mg/kg, 50mg/kg and 100mg/kg) or 5-Fu(25mg/kg) once every other day. Tumor weigh and size were measured to evaluate the anti-tumor effect of the mycelium of Taiwanofungus camphoratus in vivo. AS presented in Table 3, tumor weighs in mycelium treatment group were obviously decreased relative to the negative control group, and the tumor inhibitory rates reached 15.74% and 37.36% in 12.5 and 100 ug/mL mycelium treatment group respectively. It is well known that the drug amount of the animal experiment is large, the amount of ADC isolated is not enough to carry out aniaml experiment. So I'm sorry to be able to provide only preliminary results of anti-tumor effcet of mycelium in vivo.

Table 3 The antitumor effect of the mycelium of Taiwanofungus camphoratus on A549 cells tumor-burdened mice

Sample

Dose

mg/kg

Dosage Regimen

Number of animal(Initial/Final)

Weight(Initial/Final)(g)

(Initial/Final)

Tumor weight(g)(X±SD)

InhibitionRate(%)

Negative Control

/

ig

15/15

18.7/25.5

3.24±0.55

Mycelium of Taiwanofungus camphoratus

12.5

ip

10/10

18.6/24.6

2.73±0.42

15.74

100

ip

10/10

18.7/24.1

1.71±0.47

37.36

5-Fu

25

ip

10/10

18.5/23.2

0.78±0.27

54.39

6. Several typographical errors were noted throughout the manuscript and also even in the abstract

Response: We have consulted with an English language editing company to improve the manuscript. PDF proofs have also been uploaded as a supplemental document.
